# An EC-Earth coupled atmosphere-ocean single-column model (AOSCM.v1_EC-Earth3) for studying coupled marine and polar processes

Kerstin Hartung [1,2,3], Gunilla Svensson [1,2,3], Hamish Struthers [4,5], Anna-Lena Deppenmeier [6], and Wilco Hazeleger [6,7]

[1]Department of Meteorology, Stockholm University, Sweden

[2]Bolin Centre for Climate Research, Stockholm University, Sweden

[3]Swedish e-Science Research Centre, Sweden

[4]NSC, Linköping, Sweden

[5]Linköping University, Sweden

[6]Wageningen University, The Netherlands

[7]Netherlands eScience Center, The Netherlands

**Correspondence:** Kerstin Hartung (kerstin.hartung@misu.su.se)

**Abstract.** Single-column models (SCM) have been used as tools to help develop numerical weather prediction and global climate models for several decades. SCMs decouple small-scale processes from large-scale forcing, which allows the testing of physical parametrizations in a controlled environment with reduced computational cost. Typically, either the ocean, sea-ice or atmosphere is fully modelled and assumptions have to be made regarding the boundary conditions from other subsystems, adding a potential source of error. Here, we present a fully coupled atmosphere-ocean SCM (AOSCM), which is based on the global climate model EC-Earth3. The initial configuration of the AOSCM consists of: NEMO3.6 (ocean), LIM3 (sea-ice), OpenIFS cycle 40r1 (atmosphere), and OASIS3-MCT (coupler).

Results from the AOSCM are presented at three locations: the tropical Atlantic, the midlatitude Pacific, and the Arctic. At all three locations, in-situ observations are available for comparison. We find that the coupled AOSCM can capture the observed atmospheric and oceanic evolution based on comparisons with buoy data, soundings, and ship based observations. The model evolution is sensitive to the initial conditions and forcing data imposed on the column. Comparing coupled and uncoupled configurations of the model can help disentangle model feedbacks. We demonstrate that the AOSCM in the current setup is a valuable tool to advance our understanding in marine and polar boundary layer processes and the interactions between the individual components of the system (atmosphere, sea-ice and ocean).

# 1 Introduction

Single-column models (SCM) have been used for several decades to advance our understanding of physical processes and their parametrizations in numerical models. SCMs originated from bulk models (Kraus and Turner, 1967; Niiler and Kraus, 1977). The first vertically resolved SCMs were developed in the late 1980s. For example, Betts and Miller (1986) demonstrated added value of an atmospheric SCM framework for the development and evaluation of a convective adjustment scheme in atmospheric models, and Price et al. (1986) used an ocean SCM to study the diurnal cycle of the mixed layer in the subtropical Pacific. Research with SCMs is a valuable addition to studies with three-dimensional numerical weather prediction (NWP) models and global climate models (GCM). By zooming into a single grid column of a host model, either in the atmosphere, the ocean, or the sea-ice, one achieves a separation between resolved large-scale processes and processes parametrized in the vertical column. This means that physical processes, and the ability of their associated parametrization schemes to produce the correct physical tendencies, can be studied in a controlled framework (Randall et al., 1996). Similar to the setup of a three-dimensional model, initial conditions are provided, typically from a sounding, mooring or a reanalysis profile. Although the column is decoupled from the large-scale flow, forcing mimicking the influence of the large-scale circulation on the column of interest can be applied. In practice, this is done by applying pressure gradient forcing via the geostrophic wind, horizontal advection, and vertical velocity forcing to the atmospheric component of the SCM. Relaxation (nudging) is an alternative way to include forcing by the large-scale environment. Forcing types can also be applied in combination, depending on the type of model experiment being performed. In the controlled environment of an SCM, the evolution of idealized or realistic initial profiles exposed to forcing of varying complexity can be studied in an Eulerian or Lagrangian setting. The choice of experimental setup determines how, and to what extent, different physical parametrizations within the model can be studied. Thus, an experiment needs to be designed carefully, depending on the underlying scientific question. By only evolving a single grid column, the computational cost is reduced considerably compared to experiments with a three-dimensional model. This allows for comprehensive parameter testing as more sensitivity experiments can be carried out. In summary, an SCM can be a powerful tool if its limitations are handled with care.

For these reasons, SCMs have regularly been employed to investigate modelling of physical processes in the ocean, sea-ice and atmosphere. In the ocean, single-column models, sometimes just called column models, started off as bulk mixed-layer models (Kraus and Turner, 1967; Price et al., 1986). From the start, they were used to study the impact of air-sea exchange and vertical mixing on the temporal evolution of the oceanic mixed-layer. In Gaspar et al. (1990) and Large et al. (1994), these bulk models are extended to 1D turbulence models which can be applied in the whole column and are thus suitable for GCMs. More recent examples of oceanic SCM models being used for model development are Ling et al. (2015) and Reffray et al. (2015).

In addition to research with individual atmospheric SCMs (e.g. Betts and Miller, 1986; Randall et al., 1996), SCM intercomparison studies have focused on e.g. convection (e.g. Betts and Miller, 1986; Ghan et al., 2000; Bechtold et al., 2000; Lenderink et al., 2004), stratocumulus (e.g. Bretherton et al., 1999; de Roode et al., 2016), mixed-phase clouds (e.g. Klein et al., 2009; Pithan et al., 2016), and the representation of the boundary layer (e.g. Cuxart et al., 2006; Baas et al., 2010; Svensson et al., 2011, as part of GABLS (GEWEX Atmospheric Boundary Layer Study, Holtslag, 2006; GEWEX: Global Energy and Water

EXchanges). These studies also present a wide range of numerical approaches to initialize (e.g. idealized or based on measurements), and force the model (e.g. Eulerian or Lagrangian). Idealized model setups are commonly complemented by large eddy simulations (LES) or cloud-resolving models (CRM), capturing the atmospheric evolution in more detail. LES and CRM are used to compile forcing data or as benchmarks when evaluating the performance of parametrizations in SCMs (e.g. Bechtold et al., 2000; Guichard et al., 2004; Beare et al., 2006). The cases developed within GCSS (GEWEX Cloud System Study) and GABLS, which merged into GASS (GEWEX Global Atmospheric System Study) at the end of 2010, have been successfully used to identify and improve parametrized processes (e.g. Lenderink et al., 2004) and serve as testbeds for model development. 44% of modelling centres, which develop coupled atmosphere and ocean models, polled by Hourdin et al. (2017) reported the use of SCMs for model development and tuning. This coordinated way of working has not been, to our knowledge, as extensively utilized in the ocean or sea-ice communities.

In contrast to global climate models, SCMs have mostly been implemented uncoupled. Thus, for the majority of atmospheric studies mentioned, the surface is prescribed by boundary conditions using surface temperature or fluxes. The choice of boundary condition may influence the results. Using prescribed surface temperature has proven to lead to very different energy content in the boundary layer (Svensson et al., 2011), while using different land models also introduces spread (Bosveld et al., 2014 and GABLS4), a subject that is currently further studied in DICE (http://appconv.metoffice.com/dice/dice.html). There are also theoretical limitations to consider, such as problems that arise when a stably stratified boundary layer is forced with surface fluxes (Basu et al., 2008). Over sea-ice, the presence of snow modulates the surface energy budget and thus results vary depending on the description of snow in the surface model (Pithan et al., 2016). In the ocean, the depth of the mixed layer is sensitive to the coupling, especially in the tropics and during summer, when the mixed layer is shallow and quickly responding to forcing. The fast response can give rise to positive feedbacks between model biases in the atmospheric and oceanic mixed layers (Breugem et al., 2008; Toniazzo and Woolnough, 2014). It is common to develop model components using prescribed forcing, i.e., ocean and land models use near-surface observed or reanalysis mean state variables to provide atmospheric fluxes. However, this can lead to surprises when model components are interactively coupled. Atmospheric models are forced with observed sea surface temperatures (SST) over the ocean and often developed in a framework with an interactive land model over land, although the land model is taken as is and i.e. thus not developed in the interactive framework. To avoid ambiguities arising from specifications of surface boundary conditions, it is desirable to combine several SCMs into one coupled model, especially when studying boundary layer processes or processes that depend on interfacial coupling.

In the last two decades a few coupled single-column models have been developed. Clayson and Chen (2002) coupled an atmosphere and an ocean SCM to study tropical atmosphere-ocean feedbacks, and Goyette and Perroud (2012) combined a 1D lake model with an atmospheric column model. More recently, West et al. (2016) coupled a one-dimensional sea-ice and an atmospheric column model to investigate the optimal interface at which to calculate the surface energy budget.

Following this line of work, we present a coupled atmosphere-ocean sea-ice SCM (AOSCM) following the global climate host model EC-Earth (Hazeleger et al., 2010, 2012). The AOSCM provides a platform to study both physical and numerical coupling processes at the surface interface. First, we present and discuss ways to set up and force the model. This encompasses idealized and realistic initial conditions and forcing, Eulerian and Lagrangian setups, short-term case based or long-term

statistical analysis. Application of the AOSCM is demonstrated at three locations, namely mid-latitudes, tropics and the Arctic. Varying experimental designs display the versatility of the tool.

## 2   Model description, model setup and data

### 2.1   Model components

In this study, the AOSCM is build from the atmospheric model OpenIFS (Open Integrated Forecasting System, https://software. ecmwf.int/wiki/display/OIFS/Single+column+model+40r1+release+notes), including the land model H-Tessel (Balsamo et al., 2009), and the ocean model NEMO (Nucleus for European Modelling of the Ocean, https://www.nemo-ocean.eu/) with the sea-ice model LIM (Louvain-la-Neuve Sea Ice Model, http://www.elic.ucl.ac.be/repomodx/lim/). All coupling actions between the column versions of the sub-components NEMO and OpenIFS are performed by the coupling software OASIS3-MCT

(https://portal.enes.org/oasis). For model development purposes, the column model should follow the specifications of a GCM host model. In an iterative process, findings from the SCM, and specifically their impact on the large-scale circulation, can then be directly tested and evaluated in the GCM. In this way the computational cost for coupled model development is reduced. Here, the AOSCM is set up to closely match the development version of the EC-Earth model. Presently, this means that the default setup is a column version of EC-Earth v3, except that instead of using IFS cycle 36r4 the AOSCM uses OpenIFS

cycle 40r1. Future versions of EC-Earth will be based on OpenIFS. The other components, namely NEMO3.6, LIM3 and OASIS3-MCT, are used with the same version in both EC-Earth v3 and the AOSCM.

The different model components are presented with a focus on formulations and settings specific to the one-dimensional versions of the codes. Still, the description does not encompass all details on the model subcomponents. This is mainly motivated by the fact the AOSCM, as well as all its components, are continuously under development. For current settings and

recent updates we refer to the AOSCM code branch and the respective model platforms.

### 2.1.1   OpenIFS

OpenIFS (hereafter OIFS) is developed by the European Centre for Medium-Range Weather Forceasts (ECMWF) as a version of IFS intended for research and education (Day et al., 2017). The main difference of OIFS40r1 to IFS 40r1 is the exclusion of the data assimilation component of IFS. Extensive documentation is available for IFS at: https://www.ecmwf.int/en/forecasts/

documentation-and-support/changes-ecmwf-model/cycle-40r1/cycle-40r1.

The atmospheric part of the AOSCM solves the one-dimensional version of the primitive equations:

$$-\dot{\eta}\frac{\partial u}{\partial \eta} + F_u + f(v-v_g) + P_u + \frac{u_r - u}{\tau_a} = \frac{\partial u}{\partial t} \tag{1}$$

$$-\dot{\eta}\frac{\partial v}{\partial \eta} + F_v - f(u-u_g) + P_v + \frac{v_r - v}{\tau_a} = \frac{\partial v}{\partial t} \tag{2}$$

$$-\dot{\eta}\frac{\partial T}{\partial \eta} + F_T + \frac{RT\omega}{c_p p} + \quad P_T + \frac{T_r - T}{\tau_a} = \frac{\partial T}{\partial t} \tag{3}$$

$$-\dot{\eta}\frac{\partial q}{\partial \eta} + F_q + \qquad P_q + \frac{q_r - q}{\tau_a} = \frac{\partial q}{\partial t} \tag{4}$$

As in the full model system, a two-time-level semi-Lagrangian scheme is used (an Eulerian scheme is also available) to integrate the momentum with horizontal wind components $u$ and $v$ (Eq. (1) and (2)), thermodynamics $T$ (Eq. (3)), moisture $q$ (Eq. (4)) as well as the continuity equation. The vertical coordinate is based on $\eta$ levels, which merge orography following $\sigma$ coordinates near the surface with pressure coordinates in the free atmosphere. Here, $\dot{\eta}$ and $\omega$ are vertical velocities, in $\eta$ and pressure coordinates, respectively. $F_i$ is the horizontal advection, $P_i$ summarizes physical parametrizations and $u_r, v_r, T_r, q_r$

denote the reference profiles used for nudging with a time scale $\tau_a$. Furthermore, $f$ is the Coriolis parameter, $u_g$ and $v_g$ the horizontal components of the geostrophic wind, $R$ the moist air gas constant, $c_p$ the heat capacity of moist air at constant pressure and $p$ the pressure. In addition to the atmospheric state variables (Eq. (1) - (4)), the model prognostically calculates cloud liquid, ice, rain, snow and cloud cover.

The total tendency (right-hand sides of Eq. (1) - Eq. (4)) to each prognostic variable is calculated as the sum of dynamical

(first three terms on the left-hand side) and physical parametrization tendencies $P_i$ (fourth term), possibly updated by relaxation (i.e. nudging, fifth term). The order of the left-hand side of the equation is, in a simplified way, equivalent to the sequence in which the tendencies are calculated in the model (Fig. 1). In the time-stepping loop, the dynamical tendencies are determined, mainly aggregating available prescribed forcing. The pressure gradient forcing is represented by the geostrophic wind. The third term of the heat equation captures adiabatic heating through vertical motion. Calculations of tendencies from physical

parametrizations are done in the same way as in the three-dimensional OIFS. Detailed discussion of the parametrizations used for these processes, namely, the radiation, turbulence, cloud and convection parametrization schemes as well as the non-orographic gravity wave drag, orographic gravity wave drag and surface drag, can be found in the IFS documentation for cycle 40r1 (https://www.ecmwf.int/sites/default/files/IFS_CY40R1_Part4.pdf). Relaxation tendencies are calculated weighing the difference between the new state, as determined by physical and dynamical tendencies, and a reference state, with the

relaxation timescale $\tau_a$. References states can, for example, be observed or modelled profiles of atmospheric variables. All forcing fields are read in at forcing time steps and linearly interpolated at intermediate model steps.

Besides visualising the sequence of main routines called during an OIFS SCM run, Fig. 1 also highlights in red communications with other AOSCM components through the coupler, and use of coupling variables. Coupling variables are also

schematically shown in Figure 4. They enter the primitive equation system (Eq. (1) - (4)) via the surface energy budget (Eq. (5)).

$$(1 - \alpha_i)(1 - f_{Rs,i})R_s + R_T - \epsilon\sigma T_{sk,i}^4 + H_i + LH_i = Q_T = \Lambda_{sk,i}(T_{sk,i} - T_1) \tag{5}$$

The energy budget is solved individually for each surface tile $i$, which in the coupled system are the ocean and/or sea-ice.
The downward short-wave and long-wave radiations are $R_s$ and $R_T$, with the tiled albedo $\alpha_i$, the tiled fraction of short-wave radiation absorbed at the surface $f_{Rs,i}$, the surface emissivity $\epsilon$, the Stefan-Boltzmann constant $\sigma$, the skin temperature $T_{sk,i}$, and the skin layer conductivity $\Lambda_{sk,i}$. $H_i$ is the tiled sensible heat flux and $LH_i$ the tiled latent heat flux. Upward coupling is implemented through the surface albedo and the temperature of the upper snow, sea-ice or ocean layer $T_1$.

### 2.1.2 NEMO

NEMO is based on the thermodynamics and dynamics OPA model (Océan PArallélisé) and includes the LIM3 sea-ice component. More details of NEMO can be found in Madec (2016), and Rousset et al. (2015) describes the recent version of LIM.

The ocean component NEMO3.6 is a primitive equation model based on the one-dimensional version of the Navier-Stokes equations (Eq. (6) and (7)), the hydrostatic equation, the incompressibility equation, heat and salt conservation equations (Eq. (8) and (9)), and the equation of state.

$$-\frac{\partial}{\partial z}\nu_t\frac{\partial u}{\partial z} + fv \qquad\qquad +P_u + \frac{u_r - u}{\tau_o} = \frac{\partial u}{\partial t} \tag{6}$$

$$-\frac{\partial}{\partial z}\nu_t\frac{\partial v}{\partial z} - fu \qquad\qquad +P_v + \frac{v_r - v}{\tau_o} = \frac{\partial v}{\partial t} \tag{7}$$

$$-\frac{\partial}{\partial z}K_t\frac{\partial T}{\partial z} + \frac{1}{\rho_o c_p}\frac{\partial I(F_{sol}, z)}{\partial z} + P_T + \frac{T_r - T}{\tau_o} = \frac{\partial T}{\partial t} \tag{8}$$

$$-\frac{\partial}{\partial z}K_t\frac{\partial S}{\partial z} + E - P \qquad\quad +P_S + \frac{S_r - S}{\tau_o} = \frac{\partial S}{\partial t} \tag{9}$$

EC-Earth v3 uses an equation of state which is based on conservative state variables and provides better conservation constraints than other representations of the equation of state (polyTEOS10-bsq, IOC and IAPSO, 2010). That is of less importance in the 1D version, which is therefore based on a simpler equation of state (polyEOS80-bsq, Fofonoff and Millard, 1983). The prognostic variables of the EOS used in the 1D version are the tracers potential temperature $T$, practical salinity $S$, and the horizontal velocity components $u$ and $v$ as described in Eq. (6)-(9). Here, $\nu_t$ and $K_t$ are the vertical turbulent viscosity and diffusivity, respectively. $I(F_{sol}, z)$ denotes the penetrative part of the solar surface heat flux, and $E$ and $P$ are the evaporation and precipitation fluxes. $P_i$ summarize physical parametrizations and $u_r, v_r, T_r, S_r$ again describe reference profiles to which the modelled profiles can be relaxed with a time scale $\tau_o$. The terms on the left hand sides of the equation system capture the column forcing.

The general structure and work-flow in the NEMO and LIM models are summarized in Fig. 2 and 3. The main ocean integration is organized from the time stepping routine (*stp_c1d*), with tracer and momentum tendencies evaluated separately.

The AOSCM setting includes physical parametrizations $P_i$, for example describing the turbulence closure. In the standard setting, the vertical mixing scheme is based on a TKE dependent eddy coefficient and a 1.5 turbulent closure for convection but other turbulence schemes are implemented in the code and can easily be selected. A Langmuir circulation parametrization is also turned on and the effect of chlorophyll on heating due to solar penetration is taken into account. Advection of tracers is not possible in the one-dimensional framework but can, in a similar way to that applied in the atmospheric model, be approximated by relaxing profiles of both tracer and momentum fields towards reference profiles. However, this procedure is not utilized in the examples presented here.

Communications with other components during the work-flow are highlighted in red (Fig. 2). Coupling actions are performed at the beginning of the time stepping, namely receiving fields as part of the boundary condition routines, and at the end of the time stepping, when the updated SST and ice parameters are sent to the atmospheric part of the AOSCM. The surface boundary conditions for the momentum and tracer variables are given in Eq. (10) - (13). There, $\tau_{u,v}$ are the surface wind stress components, $\rho_0$ is the in situ density, and $S_t$ the rate of change of the sea-ice thickness budget. Only the non-penetrative part of the net surface heat flux (see Eq. (5)) influences the temperature boundary condition.

$$\nu_t \frac{\partial u}{\partial z} = \frac{\tau_u}{\rho_0} \tag{10}$$

$$\nu_t \frac{\partial v}{\partial z} = \frac{\tau_v}{\rho_0} \tag{11}$$

$$K_t \frac{\partial T}{\partial z} = \frac{Q_T}{\rho_0 C_p} \tag{12}$$

$$K_t \frac{\partial S}{\partial z} = \frac{(E - P - S_t) \cdot S(z=0)}{\rho_0} \tag{13}$$

LIM3, the sea-ice model embedded in the oceanic component of the AOSCM, contains a thermodynamic and a dynamic component. In its 1D version, only the thermodynamic model is currently used, including the representation of subgrid-scale distributions of ice thickness, enthalphy, salinity and age. The model includes multiple sea-ice categories of different ice thickness, set to five categories as a default. The distribution of sea-ice thickness categories is determined based on the mean ice thickness and is constant in time. The sea-ice concentration in each category varies due to source and sink processes of sea-ice. The halo-thermodynamics parametrized in the model are solved for each ice category, which consist of one snow layer and potentially several ice layers. A brief description of the model sub-components is given in Fig. 3.

### 2.1.3 OASIS3-MCT

The OASIS3-MCT coupler (Valcke, 2006) takes care of communications between the atmosphere and the ocean/sea-ice components, and carries out transfers and temporal transformations of variables. Regridding is not necessary since two SCMs are coupled. Coupling between the atmospheric and oceanic models is performed by OASIS writing (*oasis_put*) and reading (*oasis_get*) actions (see Fig. 1 and 2). At every coupling step (a multiple of each model's time step), coupling variables are exchanged between the components. It is recommended to use a temporal lag between OASIS writing and reading actions to

avoid long-waiting times of components or possible deadlocks, even in a single-column setup. In this framework, the variables are written a given time before the coupling timestep, usually determined by the model timestep, but only read by the receiving model at the coupling timestep. Thus, initialization files of the coupling variables are needed at the start of the simulation.

Variable transfer between NEMO and OIFS is implemented in both directions (Fig. 4). NEMO receives from OIFS surface stress, solar radiation, longwave radiation, sensible and latent heat fluxes, the temperature sensitivity of the non-solar heat fluxes (long-wave radiation, sensible and latent heat flux), precipitation, and evaporation. In the reverse direction, only the sea surface temperature is passed in ice-free conditions. In presence of sea-ice, sea-ice albedo, thickness, fraction (areal coverage), temperature, and snow thickness are also transferred from LIM to OIFS. Sea-ice parameters are available for the different sea-ice thickness categories but the aggregated mean is transferred to the atmosphere. If sea-ice is present, some ice parameters are also coupled to the ocean model. The ocean receives, in addition to the atmospheric parameters, sea-ice fraction, thickness, temperature, and albedo. The rate of change in ice thickness is added to the mass flux received from the atmosphere, evapora-tion, and precipitation. OASIS3-MCT allows to pass either instantaneous values of the coupling fields at the time of coupling, or transform the field by calculating an average, maximum, minimum or sum over the period since the last coupling. As in EC-Earth v3, coupling parameters are averaged over the coupling timestep.

## 3  How to design an (AO)SCM experiment

As mentioned in Sec 1, the freedom in setting up the model initial conditions and forcing is both an advantage and a challenge when using the AOSCM. One needs to find a balance of forcing settings, based on the research question to be studied. Here, we briefly present some possibilities of using the (AO)SCM.

Figure 5 shows the main options to consider when designing an SCM experiment. Firstly, the question is if the model should be used in an idealized setting or following measurements, reanalysis, or model data. In idealized simulations, the vertical structure of initial conditions and forcing, as well as the vertical extent of the forcing, can be simplified. If no forcing is prescribed, the model column evolves in a Lagrangian way. In an SCM it would usually be assumed that the whole column is migrating simultaneously, this is unlikely to be true in reality. The Lagrangian approach of following an air parcel needs to be adapted in an AOSCM, as disregarding relative horizontal velocities of the components is unrealistic, especially for longer simulations.

More complex experiments can be designed in a variety of ways, as for example described in Randall and Cripe (1999). They are presented here in order of increasing control on the model evolution and complexity of the setup. It is often advisable to combine several of these forcing options.

Pressure-gradient forcing is one of the most basic large-scale forcings. It ensures that energy is supplied from the non-resolved large-scale pressure field to counteract energy loss through frictional dissipation near the surface. As the wind is forced to be close to the geostrophic wind, modulated by the timscale prescribed by the Coriolis parameter, it can be understood as a physically motivated relaxation. Unless nudging of the wind is applied, this forcing is necessary and it is in general advisable for longer simulations. Forcing with geostrophic winds is known to introduce inertial-type oscillations into the column (e.g.

Egger and Schmid, 1988). Advective tendencies of prognostic variables and vertical velocity also emulate the influence of neighbouring columns on the column of interest. As the vertical structure in the AOSCM might differ from the host model column or from measurements, one needs to ensure that the tendencies are physically reasonable and, if possible, prevent the model from drifting. Thus, it might be necessary to apply advective tendencies only over a specific height interval or to add relaxation forcing. It should be noted that the vertical velocity is often corrected from large-scale forcing (Sigg and Svensson, 2004), since it is a parameter not easily diagnosed in large-scale models. For example, in ERA-Interim (Dee et al., 2011) the vertical velocity is a combination of the diagnosed vertical velocity and residuals from the calculation of physical parametrizations (Nils Wedi, ECMWF, personal communication). Finally, the model column can be forced by relaxation (also called nudging). This is the forcing option which is the most dependent on the actual model state at the time the forcing is applied, and the only one which is not mimicking a process resolved in a three-dimensional model. Weighted with the characteristic time scale of relaxation, the AOSCM column mean profile is forced towards a reference profile, for example a sounding or mooring profile, or reanalysis fields. Thus, nudging can alleviate or prevent model drift, depending on the time scale chosen. Nudging best reduces biases of state variables but has been reported to lead to problems for variables describing rates, extensively documented for precipitation (e.g. Randall and Cripe, 1999; Hack and Pedretti, 2000; Ghan et al., 2000). Nudging momentum can be very helpful when evaluating cloud microphysics (e.g. Lohmann et al., 1999) but not in a study of the boundary-layer turbulence evolution. Nudging changes the equilibrium of dynamic forcing and physical parametrizations, and might mask model biases. On the other hand, nudging tendencies can be evaluated and used to diagnose model drift and imbalances. Nudging is also useful as it allows handling of inaccurate or missing information, like inertial oscillations of wind or vertical velocity forcing.

After designing initial and forcing data, the number and length of simulations needs to be decided. Measurement campaigns are usually limited in time and thus motivate shorter simulation lengths. Even if relaxation of the profile is used to prevent model drift, the impact of initial condition and forcing sensitivity might limit the model run length to which parametrizations can be evaluated.

The physical processes of interest, and the need to appropriately resolve them, determine settings of time steps, vertical grid and coupling frequency. Even though not practicable for the host model, for which settings are usually tested, it is desirable to run the SCM with highest temporal and spatial resolution. Similarly, the model can be used to develop and understand different coupling options which are less feasible in a three-dimensional model. An example of a more advanced coupling method is synchronous coupling (Lemarié et al., 2015), in which coupling fields are sent and received at the same time step.

Both pressure gradient forcing and horizontal advective tendencies are calculated based on horizontal gradients. Thus, it should be noted that when using forcing based on model data, they depend on the horizontal resolution of the host model. The resolution of the forcing is the main scale information applied in the model, apart from potential time-scale settings, which depend on the horizontal grid settings. In addition, the temporal resolution of the forcing steers how closely the observed temporal evolution can be captured.

## 4 Examples of experimental setup and evaluation

### 4.1 Experimental setup

To illustrate the versatility of the new tool, the AOSCM is applied at three different locations, namely the Pacific midlatitudes, the tropical Atlantic and the North Polar region. The locations are chosen to demonstrate the model in three different climatic regions. Result from the coupled SCM (AOSCM) are compared with atmosphere-only (ASCM) or ocean-only (OSCM) simulations.

Special focus is placed on analysing the stability of the simulations, i.e. we test for model drift, compared to gridded reanalysis data (for the Pacific midlatitudes and tropical Atlantic locations). It should be noted that evaluation against reanalysis does not assume that reanalyses present the truth. However, it allows to detect potential model drift against the forcing dataset. Simulations in the North Polar region are based on reanalysis data in a semi-idealized way, which also considers a reference LES simulation. At all locations, model simulations are evaluated against point-based observations. In addition to testing for model stability, sets of experiments at the three locations analyse the sensitivity to forcing and model settings while highlighting the versatility of the AOSCM. Furthermore, current scientific questions and avenues to study them are touched upon for two of the locations (tropical Atlantic and North Polar region). However, our aim is not to conclusively answer these science problems but to motivate other users to consider the AOSCM for such tasks.

An overview of the experiments at the three locations is given in Table 1.

Atmospheric initial conditions and forcing are obtained from ERA-Interim (Dee et al., 2011). Both analysis steps, which are provided every six hours, and intermediate 3-hourly forecast are used. The OIFS SCM is initialized with profiles of the non-cloud atmospheric prognostic variables. In case of atmosphere-only simulations, the sea surface temperature is initialized and updated daily. Restart files of surface parameters required for coupled simulations are obtained from short ASCM simulations. All forcing data, horizontal advective tendencies of the prognostic variables, geostrophic wind, and vertical velocities, are calculated from the three-dimensional fields of ERA-Interim for each output timestep.

The ocean is initialized from observed daily-mean profiles of temperature and salinity, measured to a depth of 120-500 m at the Pacific and Atlantic locations. As these depths are well below the typical mixed layers, we assume that temporally coarser data in the deeper ocean does not significantly influence the model evolution near the surface. Therefore, the observed initial profiles are extended below by monthly-mean potential temperature and salinity ORAS4 reanalysis fields (Balmaseda et al., 2013). At the Arctic location, the initial ocean profile is taken from ORAS4 data. The vertical grid is based on 75 levels, though at the Arctic location the shallow bathymetry means that only 17 levels are used. The ocean is only forced by coupling information from the atmosphere.

To ensure best performance, the equivalent resolution of the A(O)SCM is set to T511, mainly reducing the convective adjustment timescale and thereby alleviating instabilities. In contrast to EC-Earth v3, the radiation time step is equal to the dynamics time step (see Table 1). The NEMO configuration differs from the standard EC-Earth GCM settings, since it is uses NEMO-C1D options (Reffray et al., 2015). Namely, the equation of state formulation and the temporal chlorophyll structure are

adapted. Instead of a constant value, SeaWIFS based chlorophyll climatologies are used (NASA Goddard Space Flight Center, 2014). For the PAPA location, the data is the same as presented in Reffray et al. (2015). No bottom geothermal heating is parametrized and the enhanced vertical mixing schemes of EC-Earth is turned off. The timeseries of observed ocean profiles are influenced by tidal oscillations. As the model does not resolve these, the oscillations in measurements are removed by applying a running mean of 12 h (the frequency of the peak in the energy spectrum, not shown) for the comparison (Figure 6).

### 4.1.1  Midlatitudes: PAPA station, east Pacific

For the first experiment, we place the AOSCM at the PAPA mooring in the midlatitudinal north-east Pacific (nominally at $50°$ N, $145°$ W, https://www.pmel.noaa.gov/ocs/Papa). Observations at this location have been extensively used to develop physical parametrization in the ocean (e.g. Gaspar et al., 1990; Reffray et al., 2015), because the buoy is situated in a region of weak horizontal advection. Reffray et al. (2015) present a reference configuration of the NEMO column model at the PAPA mooring and test various mixing parametrizations available within NEMO.

The main experiment at the PAPA location consists of a 5-day coupled atmosphere-ocean simulation, initialized on 11 July 2014 at 18 UTC (11am local time) which is forced with 6-hourly data (AOSCM-6h). An uncoupled atmosphere-only simulations with 6-hourly atmospheric forcing (ASCM-6h) and a coupled simulation with 3-hourly atmospheric forcing (AOSCM-3h) act as sensitivity runs to the main setup. One further set of simulations highlights how model drift in the free troposphere can be minimized. Here, nudging of temperature, moisture, and horizontal wind with a timescale of $\tau_a = 6$ h above a height of 3 km is applied (AOSCM-N3km6h). In addition, the model was run with the standard setting extended by relaxing the horizontal wind with a timescale of $\tau_a = 1$ h (AOSCM-Nuv0km1h). With each of the experiment settings described above, a further sixteen 29-day long simulations started at 18 UTC on the first of the respective months (Oct 2010; Apr, Jun-Jul, Nov 2011; Mar, Aug, Nov 2012; Jun-Jul 2013; Jan, Apr, Jul-Sep, Nov 2014) are run for statistical assessment.

Surface variables are evaluated using hourly averaged PAPA mooring surface measurements. The variables used here are, with measurement error estimates in parentheses: 2 m air temperature ($\pm 0.2°$ C), SST ($\pm 0.003°$ C), 10 m wind speed ($\pm 2$ %), wind-speed corrected precipitation ($\pm 4$ mm h$^{-1}$ on 10 min filtered data with measurement threshold of 0.2 mm h$^{-1}$), long- and short-wave radiation (downwelling component with $\pm 1$ % error), and turbulent fluxes of heat.

### 4.1.2  Tropical Atlantic

The second location at which the SCM is tested, lies in the tropical Atlantic, situated at the 6°S, 8°E buoy of the PIRATA mooring array (Servain et al., 1998; Bourlès et al., 2008, https://www.pmel.noaa.gov/tao/drupal/disdel/). We choose a boreal summer month to demonstrate the AOSCM's ability to follow the SST cooling connected to the annual cold tongue development in the tropical Atlantic (Lübbecke et al., 2010; Xie and Carton, 2004). During the period of 1-30 June 2014, mooring observations of SST, radiative fluxes, and ocean temperature and salinity are available for SCM evaluation, which are complemented by ERA-Interim for the atmosphere. We perform experiments using several settings of the AOSCM and one OSCM simulation. The atmospheric column is either forced by advective tendencies and vertical velocity only (AOSCM-Jun1/12/15), or additionally, profiles of temperature, moisture, and horizontal wind are nudged above 1 km with a timescale of 6 hours

(AOSCM-N1km6h). For comparison, we also perform an ocean-only simulations (OSCM), which is forced by hourly precipitation, near-surface wind, temperature and moisture from ERA-Interim, and shortwave- and longwave radiation measured at the PIRATA buoy.

### 4.1.3 North Polar region

To explore the AOSCM in an experimental setting with idealized forcing, and to show the additional interaction with sea-ice, we choose an Arctic summer case. For this location ($76^o$ N, $160^o$ E), we have observations from the ACSE (Arctic Clouds in Summer Experiment) campaign during a warm-air advection episode in early August 2014 causing rapid ice melt (Tjernström et al., 2015). Sotiropoulou et al. (2018) use an LES to study the importance of advection for cloud evolution during this period. Here, we present results from the LES (Savre et al., 2014), in comparison with results from the ASCM, using the same

experimental setup as in Sotiropoulou et al. (2018). Furthermore, we explore the importance of coupling to the ocean/sea-ice, as well as the sensitivity to atmospheric model time step and coupling frequency, in ASCM and AOSCM experiments. With the aim to separate the influence of local and remote processes, as in Sotiropoulou et al. (2018), we turn off large-scale advection of heat and moisture.

The idealized experiment, based on simplified information from observations and reanalysis (Sotiropoulou et al., 2018),

assumes an initial ice concentration ($100\ \%$), surface albedo ($0.65$), and temperature ($273.15$ K, i.e. melting point of ice). The LES is applying a surface friction velocity of $u_* = 0.2$ m s$^{-1}$ as lower boundary condition, while it is modelled in the ASCM and AOSCM using a surface roughness, updated from its default value ($0.001$ m) to $0.06$ m to achieve approximately the same averaged $u_*$. The LES and the atmospheric component of AOSCM are initialized with the same vertical mean profiles, smoothed versions of soundings at 1 August 06 UTC, the starting time of the simulation. The atmospheric forcing consists

of a constant geostrophic wind of $5.4$ m s$^{-1}$ and advective tendencies of temperature and humidity, all derived from 6-hourly ERA-Interim data interpolated to a vertical L137 grid, but restricted vertically to the LES boundary layer height. The synoptic scale divergence (i.e. vertical advection), is not directly taken from the ERA-Interim as it generates unrealistic results. Thus, a prescribed divergence of $2.3 \cdot 10^{-5}$ s$^{-1}$ is applied over the first 18 simulated hours and then decreased by $50\ \%$, in both the LES and the SCM experiments.

### 25  4.2   Results from experiments

### 4.2.1   PAPA mooring – Case study

During 11-15 July 2014, the PAPA mooring briefly experienced an atmospheric cold advection event, followed by a period of weak advection, which was finally ended by warm advection (not shown). A cloud, which initially caps the boundary layer, rises and dissipates after about two days. Only during the last day does a cloud form again, associated with the warm advection.

AOSCM-6h reproduces the general temporal evolution as given by the forcing, but shows a mismatch in cloud height of up to 500 m, associated with temperature and moisture biases (Fig. 6 a and b). Modelled temperatures are overestimated at and below the reanalysis cloud height and are underestimated above, with cold biases peaking at the height of the modelled

cloud. In addition, the AOSCM produces too much water vapour mixing ratio relative to ERA-Interim. In the reanalysis, the cloud dissipates during 13 July, whereas at least a thin cloud is persisting for most of the simulation time in the three model experiments. The atmospheric boundary layer height varies around a depth of 500 m and the oceanic turbocline stays shallow, reaching at most 20 m (Fig. 6c). Atmospheric evolution and biases are similar in AOSCM-3h and ASCM-6h. During a period of weak atmospheric advection, the frequency with which forcing information is updated is thus not influencing the evolution of the coupled column.

Figure 7 summarizes the comparison between the modelled surface parameters and the PAPA measurements. If the model forcing is updated less frequently (A(O)SCM-6h), oscillations in the wind arise with larger amplitude than in AOSCM-3h (Fig. 7f). Oscillations occur mainly during periods of weak wind forcing and their amplitude increases with height (not shown). They are a sign of the column not being in geostrophic equilibrium and are enhanced if applying pressure gradient forcing, as this adds momentum to the column. At the location of the PAPA mooring, the frequency of inertial oscillations is about 16 h. A footprint of the artificial inertial oscillations is visible in the boundary layer height (Fig. 6a) and the turbulent surface fluxes (Fig. 7d, e). The flux oscillations arise from the oscillating near-surface shear, which generates turbulence. In the coupled simulations, temperature biases peak around $1°$ C (Fig. 7a, b). In ASCM-6h, a larger 2m-temperature bias can be reduced to similar values if forced with observed hourly SST instead of daily mean SST from ERA-Interim (not shown).

Comparing AOSCM-6h results to reanalysis data and PAPA measurements reveals disagreements in terms of bias signs. On the one hand, the reanalysis, and thus the forcing state, indicates that the AOSCM is too warm and moist near the surface. On the other hand, comparison to PAPA measurements points to an underestimation of atmospheric moisture (too large upward latent heat flux) and too cold near-surface temperatures. These differences might partly be explained by deviations in the SST between reality and ERA-Interim reanalysis, which steer boundary layer dynamics via stability in different ways. It is interesting to note that when the atmospheric evolution is tightly nudged to the reanalysis, the cloud structure, as well as short- and long-wave radiation improve compared to measurements (not shown). Near-surface temperature and latent heat flux however, deviate even further from observations. These differences might partly be due to compensating biases, but could also be due to non-representativeness of the buoy measurements for the model grid box. During the studied period, the AOSCM captures the local observations even with the likely erroneous large-scale forcing. Comparison with the large-scale forcing fields can be used to reveal potential atmospheric model drifts. However, in ERA-Interim the coupling to the ocean is not interactive and SSTs are only prescribed with daily resolution. One way to overcome this is to use measurements for the analysis since they reflect the observed coupling and are dependent on the true near-surface stability.

The evolution of the atmosphere is also sensitive to the initial conditions. Initializing the model only six hours later increases the biases during the final warm air advection period (not shown). In this simulation the cloud cover is underestimated, thus giving increased biases in the radiative fluxes at the surface. Furthermore, in this setup, a strong sensitivity to forcing frequency can be diagnosed, as these biases do not occur in AOSCM-3h results. Again, nudging the wind down to the surface removes the cloud biases. Initializing 18 hours earlier, on the other hand only weakly influence the results.

#### 4.2.2 PAPA mooring – Statistical assessment

Root mean square error (RMSE) statistics, relative to ERA-Interim and observations, summarize results from sixteen simulations for the main three setups AOSCM-6h, ASCM-6h and AOSCM-3h (Tables 2 and 3). Statistical significant differences are assessed by comparing the two mean values and their range of one standard deviation. If the values do not overlap considering only the range of variability from one variable we call this one-sided statistically significant. Results are separately compiled for warm and cold periods (not shown in the tables, only in Fig. 8), with eight of the sixteen simulations falling into each category. Here, warm cases are characterised by a mean ocean mixed-layer depth of less than 10 m (June-September) and cold cases by more than 30 m (November-April). Results based on oceanic profiles are not included because the variability produced by experiment setups is less than the variability among the sixteen different periods.

AOSCM-6h and ASCM-6h exhibit similar monthly mean biases in the considered parameters. Daily-mean SSTs used to force ASCM-6h simulations are one-sided statistically significantly superior to SSTs modelled by the AOSCM-6h. Reduced variability is due to a coarser temporal resolution of the forcing. The signal is largest in summer months and can be explained by SST cold biases in AOSCM runs, in some cases also present during winter. This SST bias in the AOSCM is part of a temperature bias dipole in the ocean column which intensifies with runtime. Reffray et al. (2015) discuss a sensitivity of the mixing depth to a TKE length parameter, describing the deepening of the mixed layer by near-inertial waves and ocean swell or waves. In the standard TKE setup used in EC-Earth v3, the parameter is either a function of latitude and set to 30 m at the PAPA station (standalone ocean model) or set to 0 m so that no additional mixing is supplied (coupled model). Setting the parameter to 0 m, thus not considering additional mixing by waves, produces very similar results to the ones presented here (Tables 2 and 3), but cold biases during summer months are now replaced by warm biases of roughly equal strength and too shallow mixed layers (not shown). Reducing the value of the parameter to 10 m, as suggested by Reffray et al. (2015), and thus limiting an increase of mixing depth by internal mixing, alleviates the observed summer cold biases (not shown).

In general, the AOSCM can successfully reproduce atmosphere-only results. The added benefit of a coupled simulation is that the interactions between the marine and atmospheric boundary layer are resolved and can be studied directly. AOSCM-3h, forced with atmospheric data of higher temporal frequency, is better able to represent measurements and model reference data than AOSCM-6h, with largest impact on momentum. Again the annual mean signal originates mainly from one subperiod, in this case the cold months, when AOSCM-3h performance exceeds that of AOSCM-6h in several aspects. Firstly, wind biases are statistically significantly reduced in the whole atmospheric column. Secondly, the mean column state bias is reduced, although not to an extent that is statistically significant. In addition to improvements in the mean state, an increase in the depth of the mixed layer is found in both atmosphere and ocean (not shown), related to reduced coupling biases, though again the change is not statistically significant.

Higher frequency forcing is, in many cases, linked to pronounced improvements in wind representation through reduction of oscillations in wind speed. One way of emulating this effect is to relax horizontal wind profiles in the model towards those provided by the reanalysis. Results from simulations with AOSCM-Nuv0km1h settings are summarized in the 4th column of Tables 2 and 3. Atmospheric column and surface wind biases can be reduced by nudging the wind. SST biases are also alleviated

during cold months (not shown) but atmospheric temperature and humidity biases are not sensitive to wind nudging. The ocean is affected through momentum transport during cold months. The ocean responds similarly as in AOSCM-3h simulations, though only one-sided statistically significant. The ocean mixed layer is deeper whereas the annual mean atmospheric boundary layer is shallower than in all other configurations. Thus, nudging of the wind components can be used to reduce model biases.

However, it has to be noted that wind nudging perturbs the momentum balance. Especially when studying boundary layer turbulence parametrization, nudging interferes with the performance of the parametrization.

In some simulations, the free troposphere drifts away from the reanalysis state. A weak atmospheric nudging of the four main prognostic variables temperature, moisture and horizontal wind above 3 km (i.e. well above the boundary layer, AOSCM-N3km6h) reduces biases in the troposphere even below 3 km (Table 3). At the same time, the ocean state is only weakly

influenced by deepening the ocean mixed layer. This way of nudging can be used even when the momentum balance at the surface is required to be unperturbed in the boundary layer.

Accumulated energy fluxes (see Eq. (5)) and accumulated precipitation from the main three sensitivity runs are visualized in Fig. 8, resolving individual cases. Modelled fluxes are sampled every hour to match the measurement frequency. In summary, the model surface receives too little energy during summer and loses too much energy during winter. Considering all seasons,

AOSCM-3h/6h perform best compared to ASCM-6h, but the main signal appears in different seasons. AOSCM-3h gives the best net surface energy balance during summer, and during winter AOSCM-6h exceeds the other setups. However, the overall variability is large and individual cases may show different results. Precipitation is larger during winter and the model produces generally more rain than observed.

### 4.2.3 Tropical Atlantic

Our second marine test location is the tropical Atlantic. During the time of the case study, June 2014, SSTs in this area cool by 4 °C. This trend is part of the cooling of the eastern tropical Atlantic due to its annual cycle (Lübbecke et al., 2010; Xie and Carton, 2004). To estimate AOSCM performance in this region, we perform a base simulation using only advective tendencies (AOSCM-Jun1 in Fig. 9). Within ten days, two main biases develop, one atmospheric and one oceanic. Firstly, atmospheric temperatures between 0.5 km and 1.5 km are overestimated, while moisture is underestimated over the same height interval

(not shown). The patterns of these atmospheric biases are closely correlated and peak between June 14 and 17. Both biases are flow-dependent, i.e. they are not connected to a model drift but reduce again after the June 17. The RMSE in the lower 1.5 km develops similarly for temperature (Fig. 9a) and moisture (not shown). Secondly, although the cooling of the ocean surface layer is partly captured, its amplitude is underestimated, leading to a warm bias of around 2°C at the end of the simulation (Fig. 9b). It is worth noting that the ocean column follows the observations well until five days into the simulation, when the

observed ocean cooling can no longer be matched by the model. The SST bias grows, and after a short period of recovery around day 7 to 10, it increases during the course of two days and does not reduce significantly afterwards. Emergence of a model warm bias during the build up of SST cooling is a common model bias in the tropical Atlantic (Breugem et al., 2008; Toniazzo and Woolnough, 2014; Voldoire et al., 2018).

We demonstrate how the origins of the two biases can be traced back using several sensitivity experiments. Nudging above 3 km, as done in the PAPA case, also reduces the near surface bias in moisture and temperature, but in a weaker form (not shown). The atmospheric bias can largely be alleviated by nudging prognostic variables above 1 km with a time-scale of $\tau_a = 6$ h (AOSCM-N1km6h). However, the SST evolution is not influenced by atmospheric relaxation to a height of 1 km. Inspired by the indication of a flow dependent bias in the standard setup, AOSCM-Jun12 and AOSCM-Jun15 are initialized further into the period. Initializing the ocean between 12-15 June, when the largest SST bias develops, strongly improves the SST representation in the AOSCM. The atmospheric biases develop again, stronger when initialising on 12 June than 15 June.

Finally, the SST bias can be studied by decoupling the ocean from the atmosphere. This can either be done by nudging the atmospheric column strongly (e.g. $\tau_a = 0.25$ h) down to the surface (not shown) or by performing an ocean-only simulation (OSCM, Fig. 9b). Both simulations produce very similar evolutions of the SST bias (not shown). The similarities point to an oceanic origin of the SST bias, while differences to the simulation indicated 1 June indicate the impact of additional feedbacks on the bias development. Observations of the ocean current vector (available at 10 m depth during this period) indicate two maxima of about 50 cm s$^{-1}$ at 5 June and 10 June (not shown), coinciding with periods of maximum SST bias in all simulations initialized at 1 June. The ocean model currently does not capture horizontal temperature advection. Temperature changes related to advection hence cannot be reproduced by the OSCM. Heat budget analyses shows these terms to be small in the region of the experiment (Giordani et al., 2013; Deppenmeier et al., 2018). However, short time scale events are likely to be missed and can impact the budget on shorter times. Another possible oceanic origin of the bias is insufficient ocean vertical mixing of near-surface warming into the ocean. The importance of and sensitivity to vertical ocean mixing has been observed and demonstrated by Hazeleger and Haarsma (2005) and Hummels et al. (2013), among others. Too little mixing of cold water masses into the well-mixed layer as well as too little heat transport from the upper layer into the deep ocean leads to artificially warm SSTs, similar to those observed towards the end of the simulation. In the current setup, upper ocean vertical mixing only penetrates the first upper meters of the ocean column, and then stops abruptly. Replacing the relatively strongly stratified observed profile with the more gradual profile from ORAS4 deepens the mixed-layer and improves the results slightly, but still only down to 20 m (not shown). This feature and its impact on the SST evolution are currently under investigation.

### 4.2.4 North Polar region

Finally, the AOSCM is used to simulate a moist, warm air advection event in the Arctic summer. Figure 10 shows the evolution of the liquid-water content for the reference LES simulation (a) together with observational estimates of cloud top and different versions of the ASCM and AOSCM (b-f). The atmosphere-only run (b) is the most similar to the LES as it keeps a cloud with a top at about 200 m during the whole simulation. The formation of the cloud in the beginning of the simulations (not shown) is quite different. The LES initially forms a cloud with a top at about 800 m that is slowly descending under the influence of the subsidence. In all the AOSCM simulations, a cloud also forms at that height, dissipates, and after a few hours a new cloud appears with a top at around 200 m. The evolution of the simulated cloud between hour 12 and 48 is diverging from a similar state at around hour 12, with sensitivity to coupling and time-step. In a simulation with short timesteps in all model components and coupling at every time step ($\Delta t = 450$ s, Fig. 10.c), the cloud develops into a double-layered cloud at about hour 32. Using

a longer time-step, 2700 s as is used in EC-Earth (Fig. 10.d), results in a descending and thinning cloud, which at the end of the period is only present close to the surface. Returning to a shorter time-step of 900 s in the atmosphere, but keeping the ocean, ice, and coupling at 2700 s, results in a cloud that keeps its top at 200 m for a longer time (Fig. 10.e). Two simulations are run where first the temperature and then the moisture advection is turned off, the resulting cloud for the first simulation is not that

different (compare Fig. 10.e and f). When the moisture advection is removed, the cloud disappears before hour 12 (not shown).

The integrated liquid-water content between hour 12 and 48 is presented in Fig. 11. The LES liquid water path (red) varies between 50 and 150 g m$^{-2}$ during the simulation, while the observations show a wider range. Only the ASCM (blue dashed line) reaches observed values, the coupled simulations (thick lines in blue, magenta, and cyan) produce smaller liquid water paths and little variability in sensitivity tests. In this figure, the importance of advection of moisture is clearly seen (dash-dotted

cyan line, near the bottom of the figure). Without temperature advection (cyan dashed line), the cloud stays cooler and can thus hold more liquid water.

For this Arctic case, the cloud both shields the surface from the sun and increases the long-wave radiation. For the short-wave cloud effect, the surface albedo plays an important role. As discussed in Tjernström et al. (2015), the surface is changing characteristics rapidly as energy is absorbed and melting occurs. Figure 12 shows the initial albedo in the simulation (averaged

over hour 1) for the various simulations, calculated using the model's incoming and reflected shortwave radiation. The albedo during the first hour is a result of both the initialization (same for all coupled simulations) and processes changing the albedo. The albedo in the AOSCM is calculated in LIM based on the sea-ice state, and is quite different from the default albedo climatology provided to the ASCM. In the coupled simulations, the albedo spans from about 63 to 74 %, while the ASCM's albedo is at 58 %. The LES value is 65 % and constant in time. Some of these differences can be explained by how the cloud

affects the diffuse radiation and thereby the amount of reflected light at the surface. The albedo decreases over the 48 hours in all simulations, the most ($\geq 15$ %) in the simulation where the cloud disappears. This illustrates the complexity of the coupling and how these different processes influence the net energy received by the surface.

In Fig. 13, the net mean energy at the surface, with and without the sensible and latent heat flux contribution, is shown. The deviation from the dashed 1-to-1 line gives the magnitude of the turbulent fluxes. In all simulations, the turbulent fluxes present

a net source of energy for the surface i.e. stably stratified conditions dominate. However, the observational estimate (black dot) shows a small net upward flux and the overall available energy at the surface is about 40 W m$^{-2}$ less. This analysis points to differences in the vertical structure of the atmosphere.

## 4.3 Evaluation of experiments

Based on results from the PAPA station and considering atmosphere-only setups as a benchmark, the AOSCM is performing

well and is in some cases even superior to the ASCM. Extending an ASCM to an AOSCM allows us to resolve coupled processes. A sensitivity to the forcing frequency is apparent, which is largely related to deteriorated winds in simulations forced with temporally coarser data. Both the horizontal advection and the vertical wind forcing are captured more realistically with increased forcing frequency. It should be noted that a linear interpolation will result in deteriorated results even for perfect forcing data. A linear functionality is likely not a good assumption for the temporal evolution of the forcing fields.

Wind components can be nudged to alleviate oscillations in wind speed, while this process does not influence temperature and moisture evolution. Nudging wind down to the surface ensures that atmospheric momentum biases do not deteriorate ocean performance, but the nudging interferes with parametrizations connected to momentum, e.g. turbulence. Nudging all fields above the boundary layer with weak nudging time scale remedies biases in the free troposphere while allowing to focus on the freely evolving surface interactions. At the PIRATA buoy, nudging above 3 km also reduces time-dependent atmospheric biases considerably. Biases are almost completely removed when reducing the lowest nudged height to 1 km. At the sea surface, a temperature bias remains even in an ocean-only setting or with a strongly nudged atmosphere. Both biases are sensitive to initialization time of the simulation. The sensitivity tests performed for the Arctic case, compared with both observations and an idealized LES simulation, show the complexity of how the coupling between the lower atmospheric structure, surface properties and clouds affect the energy budget at the surface. Further analysis of this case is ongoing.

Based on fluid dynamical theory and our results, we recommend to force the AOSCM with advective tendencies and pressure gradient forcing in the atmosphere. The forcing frequency should be kept as high as possible, ideally based on information from the host model at every time step, e.g. for model development. If model drifts or other persistent biases are identified, nudging profiles down to the surface, or above the processes of interest, can enhance the stability of the simulation and keep close analogies with observations. Running several sensitivity experiments based on different forcing and coupling settings, periods for further parameter sensitivity experiment can be identified and then studied inexpensively in the AOSCM.

## 5   Summary and outlook

We demonstrate a coupled atmosphere-ocean single-column model (AOSCM) following the setup of a future version of the climate model EC-Earth (v4, currently v3). The AOSCM is designed to study the physical interaction of oceanic and atmospheric boundary layer processes as well as technical aspects of the coupling. Here, we demonstrate the functionalities of the model by applying it at three locations and present analysis showing the versatility of the tool. Furthermore, we discuss possibilities of how to design process studies using the AOSCM.

As the AOSCM consists of individually compiled components, it is relatively straightforward to update and exchange routines, e.g. when newer cycles become available. At this stage, the capabilities of the EC-Earth AOSCM can be extended along several avenues. Even though the hydrometeorological variables cloud liquid, ice, rain, snow, and cloud cover are treated prognostically in OpenIFS, their profiles can currently not be forced by advective tendencies or nudging. The missing advection terms can be partially included by adding the advective tendencies of cloud liquid and ice mixing ratios to the advective tendency of water vapour mixing ratio. Similarly, advective tendency forcing could be added to both the sea-ice and ocean equation systems. Apart from the necessary model infrastructure, this requires observations or model data to compile the relevant forcing. Based on the current model without advective forcing, one option is to limit the run time of the model. In that case the relative stationarity of ocean and sea-ice with respect to atmospheric movement can be assumed. Another option is to relax the ocean profile towards a reference profile, either across the whole column or only below the mixed layer. A similar feature, namely an adaptive relaxation height, is currently not available in the atmospheric part of the AOSCM. It is, however,

possible to nudge only above a constant level, as we demonstrate. If this height is chosen well above the boundary layer, it is still ensured that the boundary layer is not affected by nudging, while biases in the free troposphere are limited.

Even though the model can be extended in numerous ways, it is a useful tool to explore several open science questions already in its current form. A non-extensive list of problems that can be tackled includes:

In the marine environment simulations similar to ASTEX (Bretherton et al., 1999; de Roode et al., 2016), describing stratocumulus to cumulus transition, can be performed with coupled models. Independent of the location where it is placed, a coupled atmosphere-ocean SCM allows to study the concept of stochastic air-sea fluxes decoupled from large-scale motions (Williams, 2012). An AOSCM, including sea-ice, presents the opportunity to study physical processes in the polar regions. The atmosphere-ice-ocean system is strongly coupled and sensitive to even small energy imbalances at the interfaces, and thus to the correct representation of sea-ice fluxes (Bourassa et al., 2013; Spengler et al., 2016). Understanding of the processes relevant for sea-ice melting and freeze-up in the changing polar regions is crucial. Work can be done along the lines of previous studies, like Pithan et al. (2016), investigating Arctic air mass transformations, and the local interactions with the snow surface (Sterk et al., 2013; Lecomte et al., 2015).

The AOSCM is a tool for investigating local, vertical interactions at the air-ice-ocean interface. All physical parametrizations implemented in the individual model components (see respective references), and their interactions, can be tested in the current setup. The AOSCM can furthermore yield insights into the physical processes responsible for model shortcomings in areas where the coupling at the interface plays an important role. With its low computational cost, it can furthermore help understand how choices of coupling parameters and numerical setup influence the evolution of the whole column.

## 6  Acknowledgements

We would like to thank for support from (O)IFS, OASIS, NEMO, and LIM communities. In addition we would like to thank ECMWF for providing ERA-Interim data and support compiling forcing information, especially Nils Wedi, Filip Vana, Glenn Carver, Irina Sandu, and Maike Ahlgrimm. In addition, we would like to thank the OCS and GTMBA Project Office of NOAA/PMEL for providing measurements at the PAPA and PIRATA buoys, the ACAS team under the lead of Michael Tjernström for the observational data and Georgia Sotiropoulou for providing the LES data. The project received support from the European Union's Horizon 2020 project APPLICATE under grant agreement No 727862 and from the EU FP7/2007-2013 under grant agreement no. 603521, project PREFACE.

## 7  Code and data availability

Usage of and access to the EC-Earth source code are licensed to affiliates of institutions which are members of the EC-Earth consortium. More information on EC-Earth is available at http://www.ec-earth.org. As the AOSCM includes the ECMWF OpenIFS single-column model, use of the AOSCM model also requires an OpenIFS license agreement, which can be obtained from ECMWF for free (https://software.ecmwf.int/wiki/display/OIFS/OpenIFS+Home).

The model source code is available from the EC-Earth development portal:

*svn checkout https://svn.ec-earth.org/ecearth3/branches/development/2016/r2740-coupled-SCM r2740-coupled-SCM*.

A tagged version on which results presented here are based can be found at *https://svn.ec-earth.org/ecearth3/tags/AOSCM.v1_EC-Earth3*. More information on the AOSCM and example data can be found on the EC-Earth AOSCM wiki page: https:

5    //dev.ec-earth.org/projects/ecearth3/wiki/Single_Column_Coupled_EC-Earth.

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

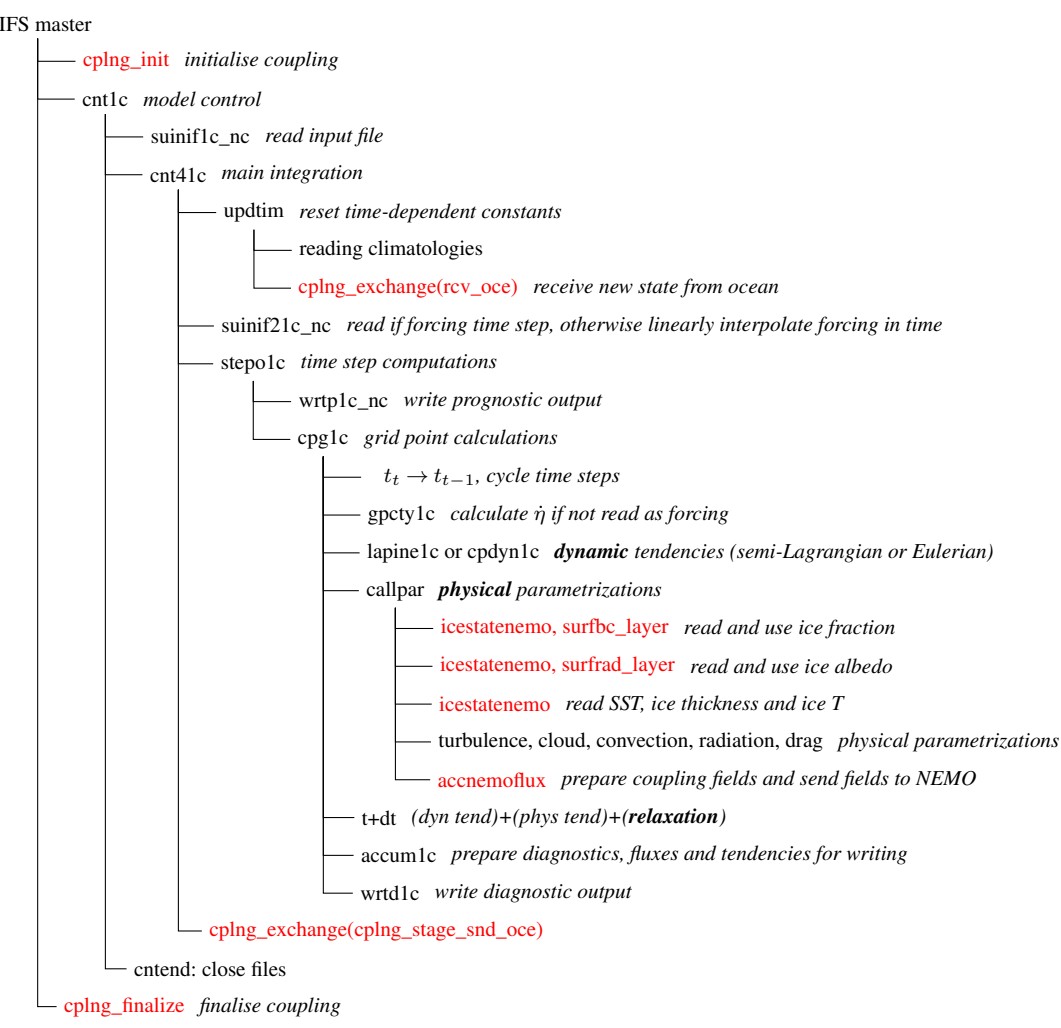

**Figure 1.** Simplified flow chart of the OIFS model. Routines dedicated to coupling via OASIS are coloured red.

NEMO, NEMOGCM
├── nemo_init  *initialise model and read namelists*
│       ├── cpl_init  *read namelists*
│       └── sbc_init  *initialize surface boundary conditions, → LIM, see Fig. 3*
├── stp_c1d  *time stepping*
│       ├── sbc  *update boundary conditions*
│       │       ├── sbc_cpl_rcv  *coupling, receiving fields*
│       │       └── sbc_ice_lim (nn_ice=3:LIM)  *update ocean surface boundary conditions, → LIM, see Fig. 3*
│       ├── zdf*  *vertical physics*
│       │       ├── zdf_tke  *TKE mixing scheme, with Langmuir parametrization*
│       │       ├── zdf_ddm  *double diffusive mixing*
│       │       └── zdf_tmx  *tidal mixing*
│       ├── dia_wri  *output dynamics and tracers*
│       ├── tra*  *advance active tracers T & S*
│       │       ├── tra_sbc  *trend due to air-sea flux and associated concentration/dilution effect*
│       │       ├── tra_qsr  *penetrative solar radiation*
│       │       ├── tra_dmp  *internal damping trends*
│       │       ├── tra_zdf  *vertical component of tracer mixing*
│       │       └── tra_nxt  *modified Leap-frog time stepping of T & S*
│       ├── dyn*  *calculate dynamics tendencies (ua: trend, ub: before, un: now)*
│       │       ├── dyn_dmp  *internal damping trends*
│       │       ├── dyn_cor_c1d  *apply Coriolis force*
│       │       ├── dyn_zdf  *vertical momentum diffusion*
│       │       └── dyn_nxt_c1d  *Euler/Leap-frog time stepping of u & v*
│       └── sbc_cpl_snd  *coupling, sending: SST, α (ice and mixed), ice fraction and thickness, sfc current*
├── nemo_closefile
└── cpl_finalize

**Figure 2.** Simplified flow chart of the NEMO model. Routines dedicated to coupling via OASIS are coloured red.

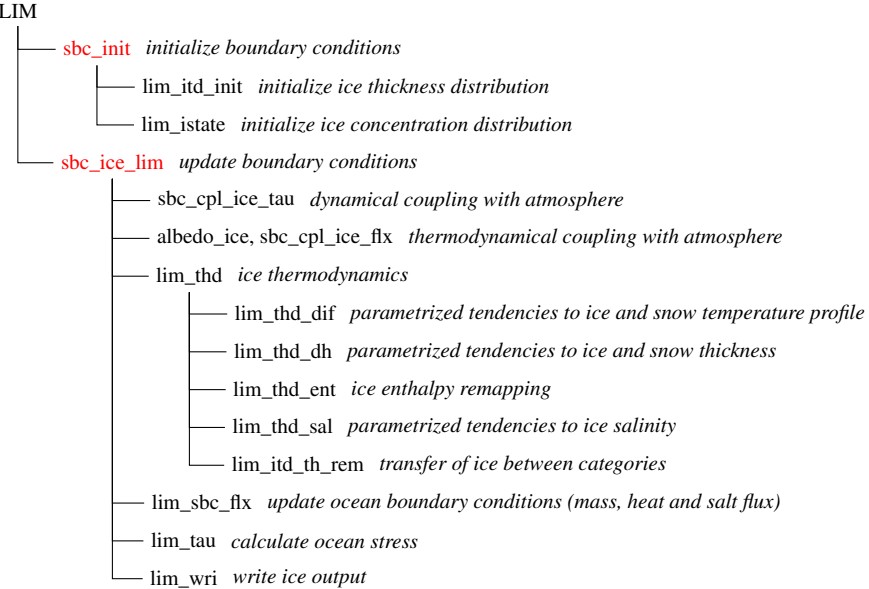

**Figure 3.** Simplified flow chart of the LIM model, part of the NEMO model if sea-ice is present. Routines dedicated to coupling via OASIS are coloured red.

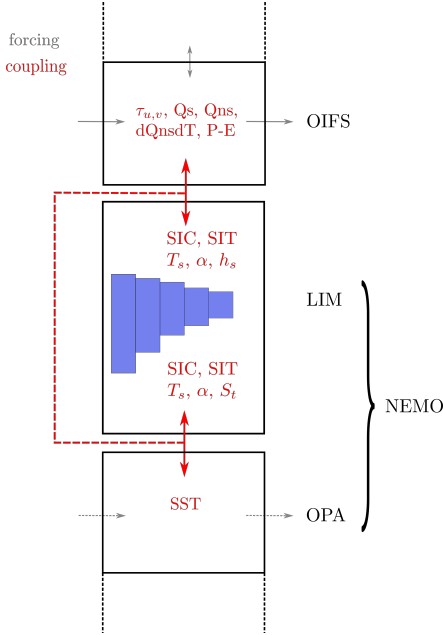

**Figure 4.** Schematic of coupling variables exchanged between the model components. In the polar environment all red lines represent the coupling (dashed and full) and without sea-ice coupling reduces to the dashed line. From the atmosphere the horizontal wind stress $\tau_{u,v}$, the solar flux $Q_s$, the non-solar fluxes $Q_{ns}$, and precipitation minus evaporation $P - E$ are passed to the ocean. In the presence of ice, the temperature sensitivity of the non-solar fluxes $dQ_{ns}dT$ is coupled as well. The ocean model sends the sea-surface temperature SST and in the presence of sea-ice the aggregated sea-ice concentration SIC, sea-ice thickness SIT, surface temperature $T_s$, surface albedo $\alpha$ and the snow thickness $h_s$. In a coupled simulation with sea-ice the ocean also receives the ice parameters SIC, SIT, $T_s$ and $\alpha$ and in addition the rate of change of the sea-ice thickness $S_t$.

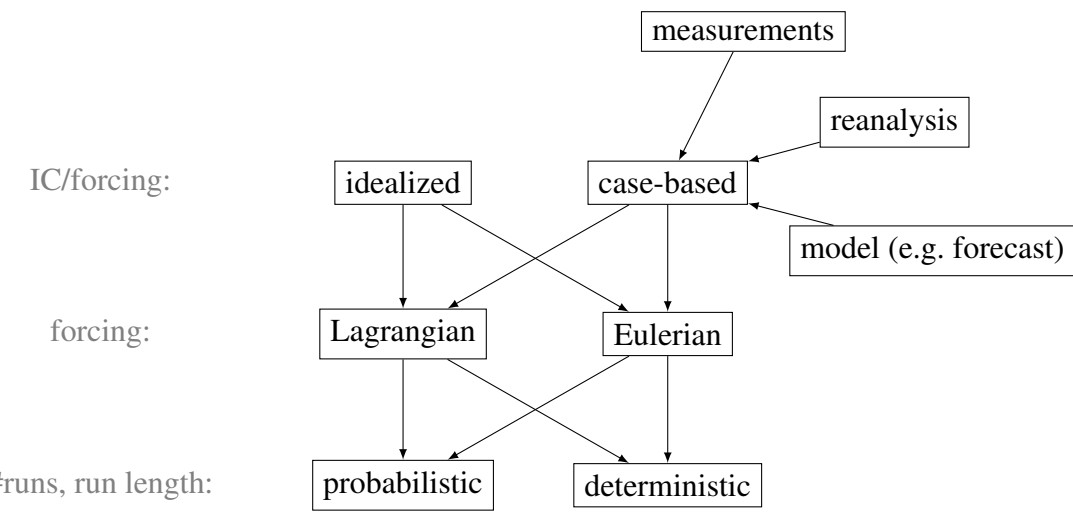

**Figure 5.** Guideline on how to set up an SCM experiment. Each row represents a setup decision necessary (grey phrase on the far left) and potential approaches. IC is short for initial conditions.

**Table 1.** Model settings at the three test locations with a selection of model parameters. Here, $\Delta t$ is the time step and $\#\text{lev}_a$ the number of atmospheric model levels. Simulations are either coupled (AOSCM), atmosphere-only (ASCM) or ocean-only (OSCM). Standard forcing includes horizontal advective tendencies, vertical velocity and geostrophic wind.

| Experiment location | Experiment type | $\Delta t$ [s] | $\#\text{lev}_a$ | Forcing | Sensitivity experiments |
|---|---|---|---|---|---|
| PAPA | AOSCM | 900 | 60 | 6h ERA-Interim | (i) ASCM, (ii) 3h ERA-Interim, (iii) nudging of uv with $\tau_a = 1$ h nudging above 3km, (iv) uvTq with $\tau_a = 6$ h |
| PIRATA | AOSCM | 900 | 60 | 3h ERA-Interim | (i-ii) initialised 12th and 15th of June instead of 1st, (iii) nudging above 1km, uvTq with $\tau_a = 6$ h, (iv) OSCM |
| Arctic | ASCM | 450 | 137 | 6h idealized (ERA-Interim and observations) | (i) AOSCM, (ii-iii) $\Delta_t \in \{900, 2700\}$ s, (iv-v) no T and q advection |

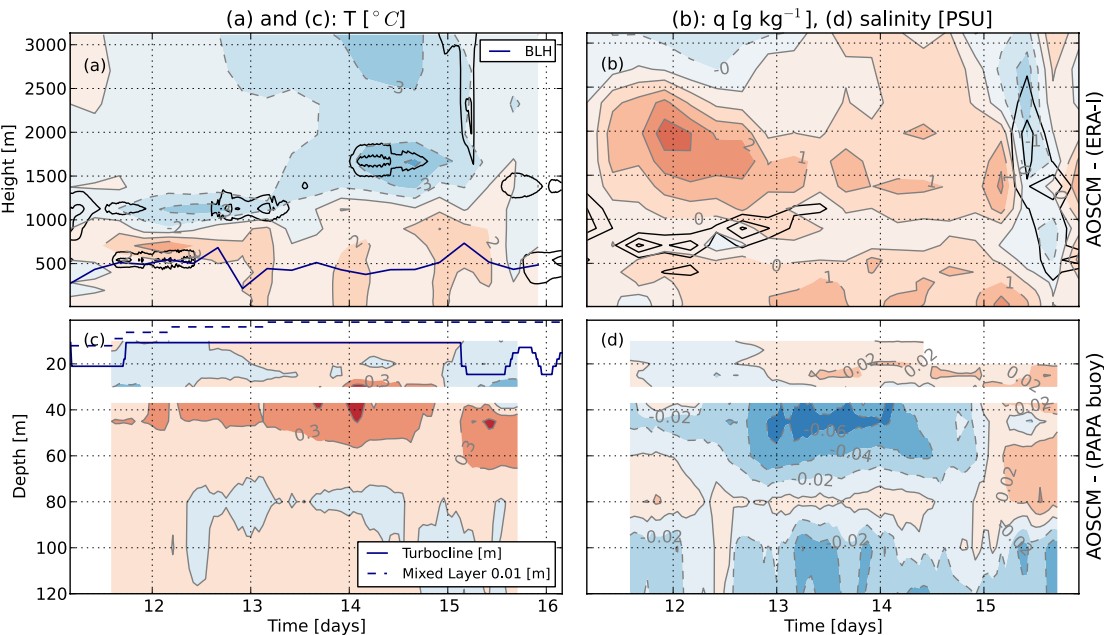

**Figure 6.** Coupled model biases of AOSCM-6h relative to ERA-Interim in the atmosphere (a and b) and PAPA buoy measurements in the ocean (c and d) for 11-15 July 2014. Note that the colour contours match different values for atmosphere and ocean. White areas indicate missing buoy data. Measured temperature and salinity evolution is smoothed with a 12 h running mean to remove tidal influences which are not explicitly modelled by the AOSCM. The liquid water content, i.e. the cloud, in the model (reanalysis) is given in panel (a) ((b)) in black contours showing 0.1, 0.2 and 0.3 g kg$^{-1}$. The boundary layer height (BLH) and the turbocline depth are calculated by the AOSCM.

**Table 2.** Surface RMSE after 28 days evaluated with respect to PAPA mooring measurements. Statistics calculated over 16 realizations of the five main experiments at the PAPA location. Table 1 describes experiments.

|  | AOSCM-6h | ASCM-6h | AOSCM-3h | AOSCM-Nuv0km1h | AOSCM-N3km6h |
|---|---|---|---|---|---|
| $T_{2m}$ [°C] | $0.9 \pm 0.2$ | $0.8 \pm 0.2$ | $0.8 \pm 0.2$ | $0.9 \pm 0.2$ | $0.8 \pm 0.2$ |
| SST [°C] | $0.6 \pm 0.3$ | $0.4 \pm 0.1$ | $0.4 \pm 0.3$ | $0.4 \pm 0.3$ | $0.4 \pm 0.2$ |
| SW rad [$Wm^{-2}$] | $84 \pm 27$ | $82 \pm 37$ | $77 \pm 34$ | $78 \pm 35$ | $77 \pm 34$ |
| LW rad [$Wm^{-2}$] | $24 \pm 5$ | $24 \pm 5$ | $23 \pm 5$ | $23 \pm 5$ | $24 \pm 4$ |
| SH flux[$Wm^{-2}$] | $13 \pm 7$ | $13 \pm 8$ | $11 \pm 5$ | $12 \pm 6$ | $12 \pm 7$ |
| LH flux [$Wm^{-2}$] | $26 \pm 13$ | $28 \pm 13$ | $22 \pm 10$ | $24 \pm 10$ | $27 \pm 13$ |
| $u_{10m}$ [$ms^{-1}$] | $2.0 \pm 0.8$ | $2.1 \pm 0.8$ | $1.5 \pm 0.3$ | $1.3 \pm 0.3$ | $1.9 \pm 0.7$ |

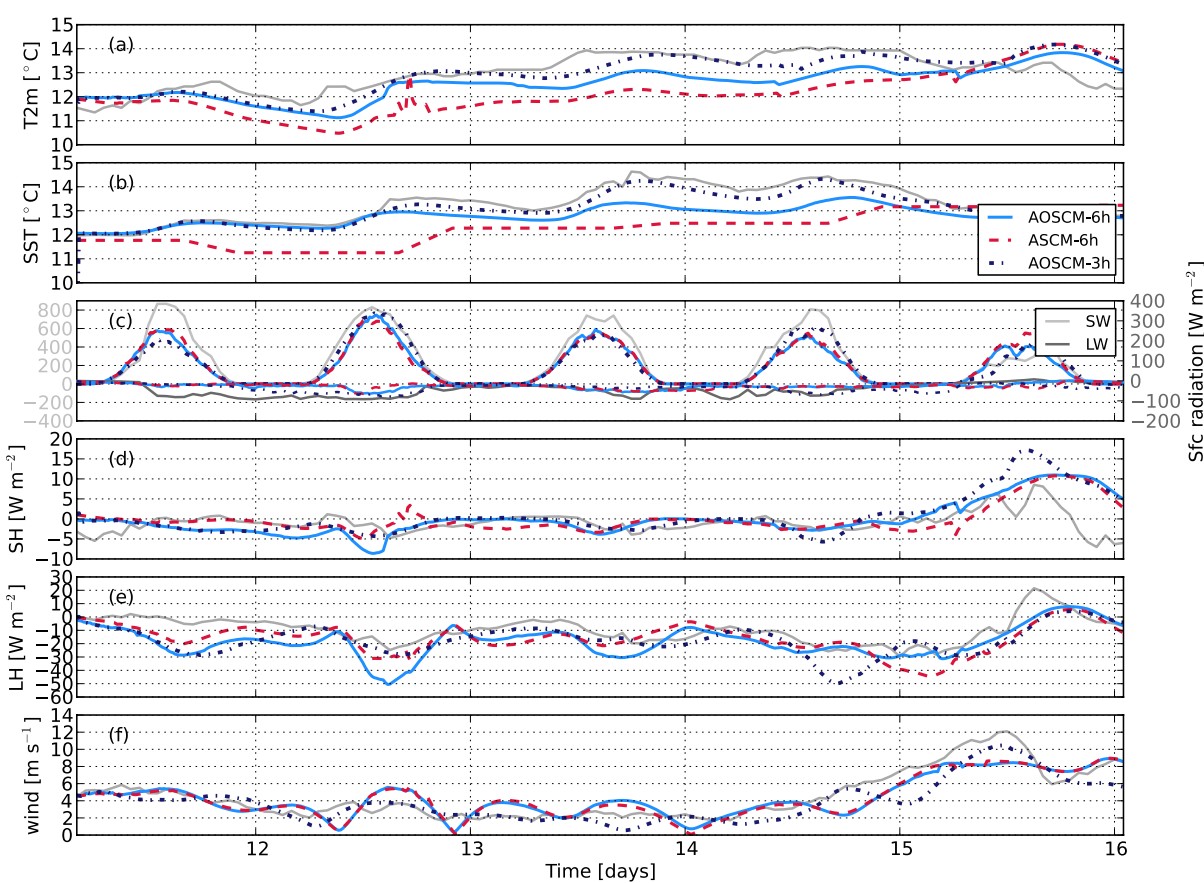

**Figure 7.** Model evolution at the PAPA buoy during 11-15 July 2014 for AOSCM-6h, ASCM-6h and AOSCM-3h. Radiative fluxes are smoothed in time with a running-mean timescale of one hour. Measurements from the PAPA buoy in grey. All fluxes are defined positive downward.

**Table 3.** As Table 2 but for RMSE of atmospheric profiles evaluated with respect to ERA-Interim fields.

| | AOSCM-6h | ASCM-6h | AOSCM-3h | AOSCM-Nuv0km1h | AOSCM-N3km6h |
|---|---|---|---|---|---|
| T [$^\circ$C], to 1km | $1.7 \pm 0.7$ | $1.6 \pm 0.6$ | $1.3 \pm 0.7$ | $1.6 \pm 0.5$ | $1.3 \pm 0.4$ |
| T [$^\circ$C], to 3km | $2.5 \pm 1.4$ | $2.5 \pm 1.4$ | $1.6 \pm 0.7$ | $2.4 \pm 1.3$ | $1.3 \pm 0.2$ |
| q [$g\ kg^{-1}$], to 1km | $7 \pm 3$ | $7 \pm 2$ | $5 \pm 3$ | $7 \pm 3$ | $6 \pm 2$ |
| q [$g\ kg^{-1}$], to 3km | $9 \pm 4$ | $10 \pm 5$ | $6 \pm 3$ | $9 \pm 4$ | $7 \pm 2$ |
| wind [$ms^{-1}$], to 1km | $3.2 \pm 1.4$ | $3.2 \pm 1.4$ | $1.8 \pm 0.5$ | $0.5 \pm 0.2$ | $2.7 \pm 1.2$ |
| wind [$ms^{-1}$], to 3km | $5.3 \pm 1.7$ | $5.3 \pm 1.7$ | $2.7 \pm 0.9$ | $0.5 \pm 0.2$ | $2.6 \pm 1.0$ |

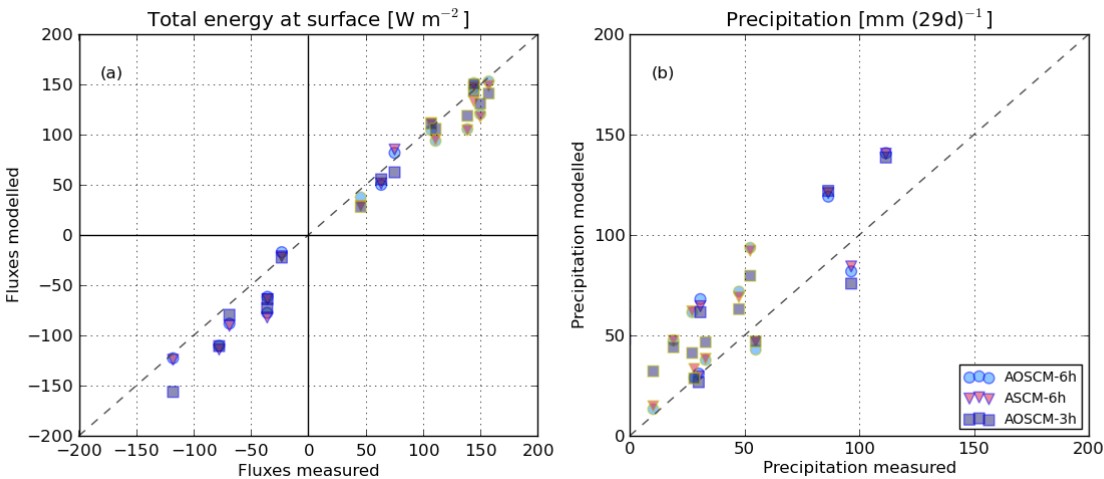

**Figure 8.** Accumulated fluxes, total surface energy and precipitation calculated over 29-day simulations at the PAPA mooring, compared for three main sensitivity runs AOSCM-6h, ASCM-6h and AOSCM-3h across all sixteen simulations. Symbols with a light (dark) border represent results from warm (cold) months. Modelled precipitation is filtered with the measurement hourly rain threshold of 0.2 mm h$^{-1}$

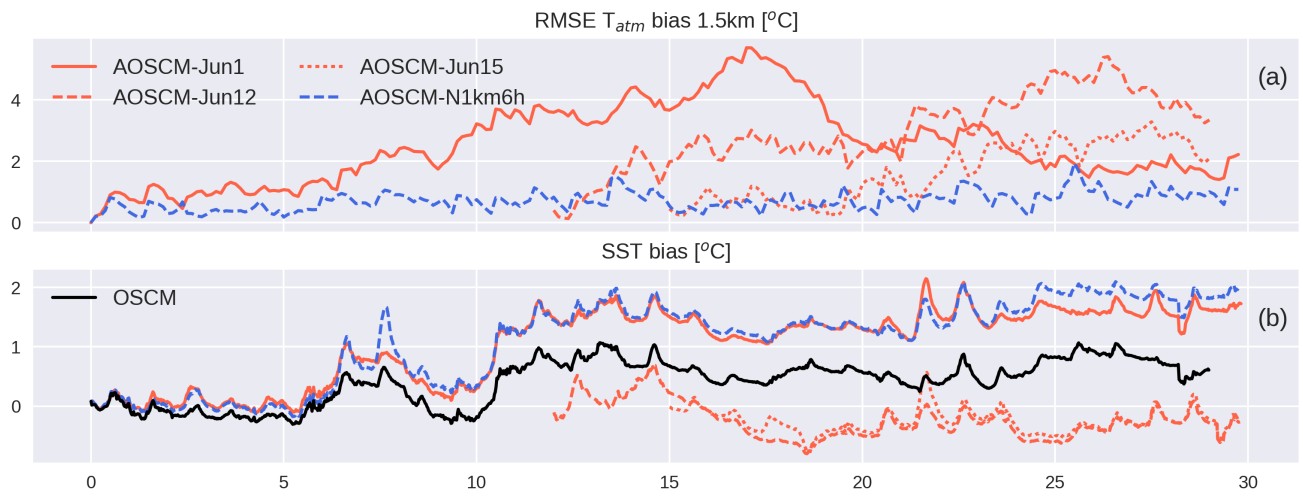

**Figure 9.** Atmospheric temperature root-mean square error integrated in lower 1.5 km of the atmosphere compared to ERA-Interim and SST biases relative to PIRATA measurements for several coupled and one ocean-only simulation. More details on the presented experiments can be found in Table 1.

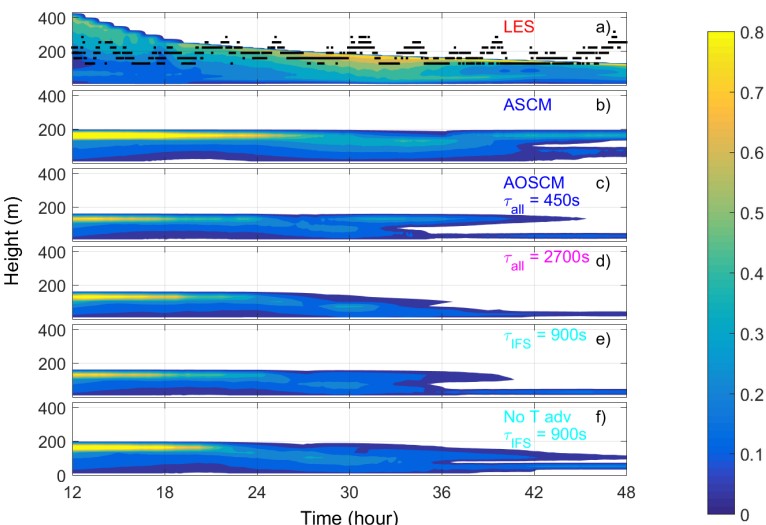

**Figure 10.** Time-height evolution of the simulated cloud liquid water content (g kg$^{-1}$) in the Arctic setup for hours 12 to 48 with a color scale that maximize at about 0.8 (g kg$^{-1}$) for a) LES results from Sotiropoulou et al. (2018), b) ASCM simulation with a time step of 450 s and 137 layers, c) AOSCM with time step 450 s in all components and coupling, d) AOSCM with conditions similar to EC-Earth i.e. 2700 s for all time steps and coupling, e) as in d) but with 900 s time step for the atmospheric component, and f) as in e) but with no temperature advection. Observational estimates of cloud top (black dots) from ACSE are also included in a).

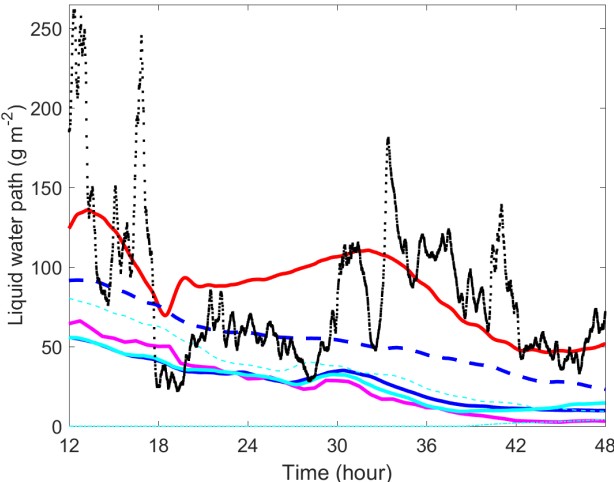

**Figure 11.** Liquid-water path in (g m$^{-2}$) for all Arctic simulations presented in Fig. 10, LES - red line, ASCM blue dashed line, AOSCM with various time steps - blue (all 450 s), magenta (all 2700 s) and cyan (IFS 900 s, other 2700 s). Also included are the results from simulations without advection of temperature (dashed cyan line) and without humidity (dash-dotted cyan thin line). Observations are shown as running averages over approximately 10 min (black dots).

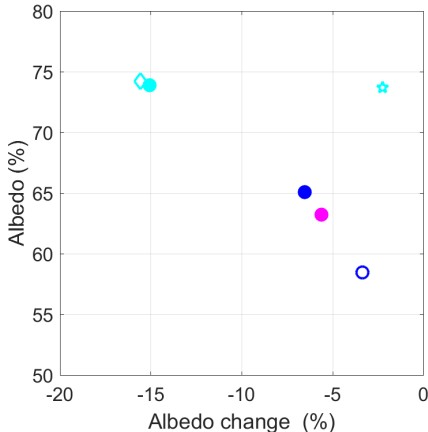

**Figure 12.** Mean albedo (%) change over the simulated 40 hours plotted against the mean albedo for the first simulated hour for the experiments in Fig. 10, same colors as in Fig. 11, ASCM open blue symbol, AOSCM simulations with no advection of temperature (cyan diamond) and no humidity advection (cyan star) are also included.

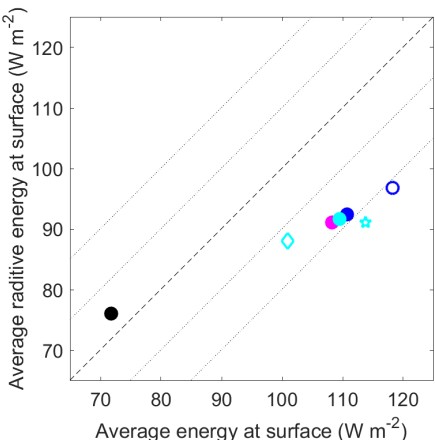

**Figure 13.** Average radiative energy as function of average energy received at the surface for hour 12 to 48 for the simulations (same symbols as in Fig. 12) and observations (black dot). The thin dotted lines around the 1-to-1 line represents $\pm 10$ and 20 W m$^{-2}$.