# Peer review of "An EC-Earth coupled atmosphere-ocean single-column model"

_Geoscientific Model Development, 2018_

## Short Comment (SC1) · 4 Apr 2018

Dear authors,

in my role as Executive editor of GMD, I would like to bring to your attention our Editorial version 1.1: http://www.geosci-model-dev.net/8/3487/2015/gmd-8-3487-2015.html This highlights some requirements of papers published in GMD, which is also available on the GMD website in the 'Manuscript Types' section: http://www.geoscientific-model-development.net/submission/manuscript_types.html In particular, please note that for your paper, the following requirement has not been met in the Discussions paper:
- "The main paper must give the model name and version number (or other unique identifier) in the title."

Please provide the version numbers of EC-Earth and AOSCM in the title of your revised manuscript.

As explained in https://www.geoscientific-model-development.net/about/manuscript_types.html GMD is encouraging authors to upload the program code of models (including relevant data sets) as supplement or make the code and data of the exact model version described in the paper accessible through a DOI (digital object identifier). In case your institution does not provide the possibility to make electronic data accessible through a DOI you may consider other providers (eg. zenodo.org of CERN) to create a DOI. Please note that in the code accessibility section you can still point the reader to how to obtain the newest version.

Yours, Astrid Kerkweg
* * *

---

## Short Comment (SC2) · 18 Apr 2018

Dear Ms Kerkweg,

thank you for your comments.

We have changed the title of our manuscript to "An EC-Earth coupled atmosphere-ocean single-column model (AOSCM.v1_EC-Earth.v3) for studying coupled marine and polar processes" and will update this in the revised version.

Usage of and access to the EC-Earth source code are licensed to affiliates of institutions which are members of the EC-Earth consortium. More information is available at

http://www.ec-earth.org. We will include this information also in the revised version of the manuscript.

Kind regards, Kerstin Hartung

---

## Author Comment (AC1) · 28 Apr 2018

We would like to note that the following information is missing from the acknowledgements in the original discussion paper:

"The project received support from the European Union's Horizon 2020 project APPLI-CATE under grant agreement No 727862 and from the EU FP7/2007-2013 under grant agreement no. 603521, project PREFACE."
* * *

---

## Referee Comment (RC1) · Anonymous Referee #1 · 29 Jun 2018

**\* General comments:**

This manuscript describes a new coupled single-column model (SCM) based on onedimensional (1D) configurations of NEMO ocean and sea-ice model, OpenIFS atmospheric model and OASIS3-MCT coupler. The technical implementation of the coupling between the models is well described and can be used as guidelines for further coupled models developments. It must be noted that SCM are extensively employed to develop and compare ocean and atmospherics models and vertical parameterizations, but independently most of the time. The originality of this work relies on the possibility to couple each component (ocean, sea-ice and atmosphere) in the same 1D framework,

and consequently to revisit and to extend the classical SCM approach. The limitations of this approach (horizontal terms are not represented in the SCM equations) are also carefully discussed and different existing solutions (prescribing lateral/vertical advection, geostrophic wind and nudging) are proposed and tested. The authors also give some useful recommendations to carefully design numerical experiments and to check the validity of the SCM results. The originality of the manuscript also relies on the multiple applications created with the coupled SCM at three different latitudes: the tropics (Pirata mooring), midlatitudes (Papa station) and polar regions (ACSE campaign). The sensitivity of the simulations to the different relaxation methods, strength and frequency is also discussed. Hence, the manuscript gives a complete description of what can be expected from a coupled SCM in realistic conditions. However, despite all these positive aspects, I found the manuscript quality uneven. This is especially true regarding the "Results" part which is poorly constructed and guite difficult to follow. I think this is mainly due to the authors intent to show a too much comprehensive study. The experimental setups and diagnostics are very different at the three locations, so no clear conclusion regarding the SCM behaviour and the relaxation methods can be easily drawn. Consequently, some work is needed to improve this disappointing part in order to get a more globally coherent and qualitative manuscript. Regarding the general form of the manuscripts, I suggest to merge the setup subsections 2.3.1 to 2.3.3 with the corresponding results subsections (3.1 to 3.3) to avoid any confusion between the three test cases. Most of the figures legends must be completed to get a better description. I also recommend a better usage of punctuation (especially commas) and a proofreading by a native English speaker to improve the manuscript readability.

\* Specific comments:

The section 2.3 structure should be improved by merging specific experimental setup sections (2.3.x) with the corresponding results section (3.x). Some experiments are named (AOSCM-3h, ...), others are not. An additional table summarising all the experiments can improve the readability of the results section. Please also address the
comments and questions in the following section (especially from p.9 to p.16).

Č \* Specific and technical corrections:

p.1 l. 11: replace "Although the model can be extended" by "Finally" p.1 l. 12: suppress "already"

p.2 I. 2: remove "already" p.2 I. 3: remove the parenthesis p.2 I. 16: "and" -> "or" p.2 I. 32: replace "and" by a comma

p.3 I.5: "GCSS" acronym is not defined p.3 I.7: "model" -> "models" p.3 I.13: "SCM studies" -> please give some references p.3 I.14: please explain why a stably stratified ABL should not be forced by surface fluxes p3. I.15: specify that this study concerns land surface p3. I.22: near-surface observations and reanalysis cannot be considered as idealized forcing, please clarify p.3 I.26: "to" -> "into" ? p.3 I.28-31: can you give more details about the main results achieved by these studies please ?

p.4 I.3: I recommend to change the title to "Model description, setups and data" p.4 I.5: "realized" -> "builded" p.4 I.10: give reference and link for OASIS3-MCT please p.4 I.27: "optional" -> "also available"

p.5 I.19: "forcing is read" -> "forcing fields are read" p.5 I.23: surface emissivity only concerns the longwave radiation emitted by the surface and not the net surface LW flux. Please correct the equation. p.5 I.26: is there a skin layer conductivity parameterization for the ocean ? If not it could be better to talk about diffusivity for the ocean instead of conductivity. p.5 I.27-28: I don't understand this sentence. Does it mean that the albedo is prognostic ?

p.6 l.1: one-dimensional version of Navier-Stokes equations p.6. l.8-9: please add a reference about equation of state formulation p.6 l.10: what is the interest to change the equation of state for the 1D ? numerical cost ? p.6 l.14: please give different time scale variable names for the different components (ocean, atmosphere, ice) p.6 l.20: please add Reffray et al. 2015 citation here. p.6 l.21: "for" -> "on" ? p.6 l.22: "way
```
similar" -> "similar way" ?
```

p.7 l.10: "communication" -> "communications" p.7 l.18: "variable" -> "variables" p.7 l.22: "transferred" -> please add "from LIM3 to OIFS with OASIS" or something similar. p.7 l.27: "means of" -> "" (useless)

p.8 l.32: "." should be "and" p.8 l.34: can you give a practical example or reference about this statement please ?

p. 9 I.14: can you give more details or practical recommendations/exemples about the relationship between the horizontal resolution of the host model and the SCM please ? p.9 I.24: the computation of the forcing data is not the same depending on the considered experiment and should consequently be moved in the corresponding experiment section. p.9 I.25: can you give explanations/practical reasons about the T511 resolution choice please ? And the convective time step p.9 I.25: "ASCM" acronym is not defined p.9 I.31: specify that ORAS4 is a reanalysis

p.10 I.4: satellite chlorophyll climatology is used for Papa only or also in the 2 other locations ? If Papa only, this sentence can be moved in the 2.3.1 section. p.10 I.21: please give the start date of the long simulation p.10 I.24-26: what about relative humidity ?

p.11 I.1: "topical" -> "tropical" p.11 I.4: "at" -> "on" p.11 I.5: if all simulations are done with 60 vertical levels, this information can be moved in the general setup section. p.11 I.20: "loosely" ? p.11 I.25: "the forcing" -> "the atmospheric forcing" p.11 I.27: what is the LES boundary layer height ? is it constant ? p.11 I.27: "All...setup." -> repetition with previous sentence p.11 I.28: why vertical advection from ERAI generates unrealistic results ? which kind of results ? please give more details about this point.

p.12 I.3: "east Pacific" -> "north-east Pacific" p.12 I.6: what about the wind conditions associated with the cold advection event ? p.12 I.14: "marine" -> "oceanic" p.12 I.15: what can you conclude from the fact that results are similar between AOSCM3h and
6h ? Please add a few comments. p.12 I.18-19: can you add in the text the local inertial period at Papa station to compare it with the simulation oscillations please ? p.12 I.27: can you add the reanalysis in Figure 7 please to facilitate the comparison ? p.12 I.30-31: you should add a figure showing this result. p.12 I.31-32: the fact that nudging improves cloud and LW but deteriorate temperature and LH suggest there is errors compensation in your simulation. This should be stated in the text.

p.13 l.1: what about the skin SST parameterization in ERAI ? p.13 l.6: "is sensitive" -> "is also sensitive" p.13 l.10: "sixteen" -> "five" ?! p.13 l.14: I think you have inverted "warm" and "cold" in this sentence. p.13 l.15: "daily-mean" -> "observed daily-mean"

p.14 l.8: "deepening" -> "deeper" p.14 l.18-19: You cannot conclude that just by looking at the surface total heat flux in Fig. 8. A surface heat budget is needed for that. p.14 l.22-23: a timeseries with observed and simulated precipitations would be more convincing than Fig. 8b.

p.15 l.5: "traced in" -> "traced back by" ? p.15 l.10: "10-15 June" -> "12-15 June" p.15 l.19: "SCM" -> "OSCM" p.15 l.24: "ocean" -> "deep ocean" p.15 l.30: "moist intrusion" -> "moist warm intrusion" p.15 l.32: "atmosphere only" -> "atmosphere-only" ? p.15 l.34 - p.16 l.1: can you explain why the cloud formation is different from the LES ? Is it because the subsidence is not represented in the model ? If yes, why not force the model with a negative vertical velocity ?

p.16 l.10: the liquid water path is integrated over the entire atmosphere height ? p.16 l.22-23: is there any observations for the surface albedo ? if yes, can you validate your model albedo ? or directly use the observed albedo in your simulations ? p.16 l.26: "all" -> "how" ?

p.17 l.2: "demonstrate" -> "describe" ? p.17 l.5-6: "highlight avenues" -> "suggest advices" ? p.17 l.8: "demonstrate" -> "present" ? p.17 l.12: "infrequent": not sure if this the correct word to describe low-frequency forcing or the problems related to inertial frequency forcing
p.18 l.27: "like thank" -> "like to thank" p.18 l.26: you recommend pressure gradient forcing without testing it directly. It would be better to present it as a promising possibility that need to be tested.

\* Figures comments:

Figure 1: please detail the acronyms such as "GWD", ....

Figure 2: "concetration" -> "concentration" "interal" -> "internal" please detail the acronyms such as "BC", "BF", ...

Figure 4: This schematic is confusing because LIM3 is a part of NEMO. Perhaps A big "NEMO" box with inside 2 small boxes such as "OPA" and "LIM3" would be easier to understand.

Table 2: a "Table 3" with oceanic RMSE would be interesting

Figure 6: - Should be moved before Tables 1-2 - A third panel showing data from ERAi and Papa mooring would greatly improve this figure. - Color bars are missing - The initial mooring data appears to be missing in the figure, how do you initialize the ocean model if so ? - Why did you chose this period if there are a lot of missing oceanic data ? - Why are mooring oceanic observations missing between the surface and -10m ? - BLH is computed from the AOSCM or ERAI ? Please clarify it. - MDL is computed from observations or the AOSCM ? Please clarify it. - (c) and (d) panels description are missing

Figure 6 I.3: "included in" -> "represented by" ?

Figure 7: - please add Q2m and precipitation timeseries - please add ERAI to check how ERAI compare with observations and the model (it will also clarify your discussion).

Figure 8: - please separate panels descriptions for (a) and (b) - legend is missing for grey squares - panel (b): wrong x axis: "fluxes" -> rain I would remove panel (b) and replace it by precipitation timeseries in Figure 6.

GMDD
Figure 9 I.1: add "temperature" to "Atmospheric RMSE error"

Figure 10 I.5: the red dots (cloud base) are not visible.

Figure 12: the empty blue circle is not described in the legend.

**GMDD**

---

## Referee Comment (RC2) · Anonymous Referee #2 · 14 Jul 2018

The manuscript provides a detailed description of a coupled atmosphere – sea ice – ocean column model and examples of its applications. The model components and the coupling methods are basically well presented. Particularly good in the manuscript is the discussion on how to design experiments applying the coupled column model. Also the Introduction is very good and motivates a reader. The manuscript has potential to become a good paper, and particularly useful for those who carry out column model experiments or plan to start such activities. The manuscript has, however, also weaknesses, and substantial revisions are needed before I can recommend it for publication.

[Figure]

Major comments

1. In several parts of the manuscript, the text is too technical, including acronyms never explained, e.g. on page 6, lines 8, 10, and 17; page 7, lines 12, 14, and 15, and many other places. This holds also for Figures 1 to 3, which are more like computer codes than illustrations that one would expect to see in a peer-reviewed journal.

2. The description of sea ice component is very brief, and leaves many open questions. How many model levels there are in sea ice and snow? How the grid is affected when ice and snow thicknesses change, i.e., is the number of levels changed or is the vertical resolution changed? Is penetration of solar radiation into snow and ice taken into account (as in case of the open ocean)? Is snow-to-ice transformation (via refreezing of melt water, rain, or sea water flooded on top of ice) taken into account? Are melt ponds taken into account? These are important questions in atmosphere – ocean coupling in Polar regions. Figure 3 does not provide any help on these issues.

3. I am concerned about conclusions made on the basis of comparisons of the model results and ERA-Interim reanalyses in the open-ocean experiments. Are the vertical profiles in ERA-Interim over remote ocean areas accurate enough to make robust conclusions on the performance of the column model? Some references or other evidence on the accuracy of the ERA-Interim profiles should be provided. Further, on page 11, line 3, it is stated that observational data are complemented by ERA-Interim for the atmosphere. It should be more clearly explained what atmospheric variables were observed.

4. The presentation of the results of the three cases studies should be clarified. It is presently hard to understand the basis of the conclusions made.

5. On page 11, line 24, it is told that 0.06 m was used for sea ice roughness. This value is much larger than observations indicate (see papers by Andreas, Lupkes and others). If the key parameter for dynamical coupling between the atmosphere and sea ice is so strongly tuned, what is the relevance of the column model experiments?

[Figure]

6. Advective tendencies were not applied in the ocean, but could be obtained from ocean reanalyses, analogously to obtaining them from ERA-Interim.

7. In Summary and Outlook the authors should better evaluate which processes can be simulated applying AOSCM and which cannot. Considering processes acting in the vertical dimension in the Arctic Ocean, there are major challenges in understanding and modelling at least the following processes: gravity drainage of salt in sea ice, brine formation, turbulent exchange of momentum, heat and salt at the ice base during ice growth, double-diffusive convection, and those mentioned in comment 2 above. After reading the manuscript, it remains unclear if AOSCM takes these processes into account.

Minor comments

Page 3, line 7: Do you mean 44% of meteorological modelling centres or modelling centres in general?

Page 5, line 18: Clarify what is meant by a reference state.

Equations (5) and (8): If part of the solar radiation penetrates below the surface, as in Eq. (8), it should not contribute to surface energy balance, but this is not reflected in Eq. (5).

Page 5, line 31: should there be a comma after "(2016)"?

Page 7, line 21: does "fraction" mean areal coverage of sea ice?

Page 8, line 23: "... only, or not, over..." is puzzling.

Page 9, line 12: Briefly explain what is asynchronous coupling.

Page 10, line 24: Does "near-surface temperature" mean 2-m air temperature?

Page 10, line 28: Was the tide only due to M2 component? Is the diurnal tide had large contribution, 12 h running means do not help. Note that tidal currents may also affect

vertical mixing in the ocean (although not necessarily much in the case presented, as it is in the deep ocean) and, if the model does not include tides, temporal averaging does not help.

Page 12, lines 29-31: Why did the results improve?

Page 14, line 31: Why not simply write RMSE instead of confusing "integrated bias".

Page 17, line 2: Why do you call EC-Earth as "the next version climate model"? It sounds very strange, particularly when referring to a paper eight years old.

---

## Author Comment (AC2) · 24 Aug 2018

We would like to thank the anonymous reviewer for the useful feedback. We have restructured the section describing the model setups and data to better motivate the choice of experiments, their similarities and differences. A table now summarises all main experiments used in this study and also gives a brief overview of sensitivity experiments. We have furthermore corrected typos and adapted suggestions for textual changes. The changes may easily be followed in the attached document containing track changes.

For suggestions requiring more explanations, we provide answers below.

**\* General comments:**

This manuscript describes a new coupled single-column model (SCM) based on one-dimensional (1D) configurations of NEMO ocean and sea-ice model, OpenIFS atmospheric model and OASIS3-MCT coupler. The technical implementation of the coupling between the models is well described and can be used as guidelines for further coupled models developments. It must be noted that SCM are extensively employed to develop and compare ocean and atmospherics models and vertical parameterizations, but independently most of the time. The originality of this work relies on the possibility to couple each component (ocean, sea-ice and atmosphere) in the same 1D framework, and consequently to revisit and to extend the classical SCM approach. The limitations of this approach (horizontal terms are not represented in the SCM equations) are also carefully discussed and different existing solutions (prescribing lateral/vertical advection, geostrophic wind and nudging) are proposed and tested. The authors also give some useful recommendations to carefully design numerical experiments and to check the validity of the SCM results. The originality of the manuscript also relies on the multiple applications created with the coupled SCM at three different latitudes: the tropics (Pirata mooring), midlatitudes (Papa station) and polar regions (ACSE campaign). The sensitivity of the simulations to the different relaxation methods, strength and frequency is also discussed. Hence, the manuscript gives a complete description of what can be expected from a coupled SCM in realistic conditions.

However, despite all these positive aspects, I found the manuscript quality uneven. This is especially true regarding the "Results" part which is poorly constructed and quite difficult to follow. I think this is mainly due to the authors intent to show a too much comprehensive study. The experimental setups and diagnostics are very different at the three locations, so no clear conclusion regarding the SCM behaviour and the relaxation methods can be easily drawn. Consequently, some work is needed to improve this disappointing part in order to get a more globally coherent and qualitative manuscript.

The main objective of our manuscript is to demonstrate the validity and the performance of the newly developed AOSCM.

To this end, we choose to run several sensitivity studies at the three locations presented. At each site, the default simulation is intended to showcase the usability of the model. This is achieved by checking that the model is not exhibiting drifts, by comparison to ocean mooring data and atmospheric reanalysis profiles.

All of the experiments are also intended to analyse potential sensitivity to forcing and model settings as well as to showcase the versatility of the tool. The latter is possible in a more natural way by presenting a broad range of experiments, even though their relationship is not immediately obvious. However, we tried to clarify similarities and differences of the simulations at the three locations.

In addition, we aim to demonstrate the usability of the model by applying it to investigate current scientific questions. This is for example done at the PIRATA location to study why global climate

models produce a warm bias in the tropical cold tongue region of the Atlantic, and at an Arctic location to study the lifecycle of mixed-phase clouds associated with the intrusion of warm and moist air masses. Even though our results can point to interesting scientific analysis it is not our goal here to comprehensively present scientific results but to highlight the application possibilities and potential.

Regarding the general form of the manuscripts, I suggest to merge the setup subsections 2.3.1 to 2.3.3 with the corresponding results subsections (3.1 to 3.3) to avoid any confusion between the three test cases.

By separating setup and results for the three locations, we can compare the motivation between the cases, highlight similarities and point to differences in the setups. The introduction to Section 2.3 is updated (now Section 4.1) to clarify similarities and differences, including the table suggested in one of the specific comments.

We now combine all experiment sections together to highlight their relation.

Most of the figures legends must be completed to get a better description. I also recommend a better usage of punctuation (especially commas) and a proofreading by a native English speaker to improve the manuscript readability.

We asked a native English speaker to check for punctuation, grammar and spelling.

\* Specific comments:

The section 2.3 structure should be improved by merging specific experimental setup sections (2.3.x) with the corresponding results section (3.x). Some experiments are named (AOSCM-3h, : : :), others are not. An additional table summarising all the experiments can improve the readability of the results section. Please also address the comments and questions in the following section (especially from p.9 to p.16).

We prefer to provide the setup and results separately as this allows us to focus more on similarities and differences in the experiments. In addition, we can motivate all three locations and the experiments performed before going into the results.

We also added Table 1 to Section 2.3 (now Section 4.1) to provide a clearer overview of the experiments. This table also gives the basic model settings used in these simulations.

\* Specific and technical corrections:

p.3 l.5: "GCSS" acronym is not defined

*Yes, "GCSS" stands for GEWEX cloud system study and we now include the definition. In addition, we have added the definition for GEWEX, which was previously missing.*

p.3 l.13: "SCM studies" -> please give some references

We have removed the ambiguous reference and restructured the text to clarify all references.

p.3 l.14: please explain why a stably stratified ABL should not be forced by surface fluxes

The paper of Basu et al (2008) describes the problem with using a prescribed sensible heat flux as a boundary conditions for a stably stratified boundary layer. Such a flux is extracting heat from the atmosphere and it can not be ensured that the atmospheric turbulence is able to keep up with the surface flux, which would lead to unrealistic development of the boundary layer. The details can be found in the referenced paper, it would be deviating too much from the subject to discuss it in detail in our manuscript.

p3. l.22: near-surface observations and reanalysis cannot be considered as idealized forcing, please clarify

This should be "using prescribed forcing".

p.3 l.28-31: can you give more details about the main results achieved by these studies please? *We find it outside the scope of this model development paper to discuss results of these studies, they are referenced mainly to acknowledge previous work along these lines.*

p.4 l.10: give reference and link for OASIS3-MCT please We have included a reference to the official website of OASIS.

p.5 l.23: surface emissivity only concerns the longwave radiation emitted by the surface and not the net surface Lwflux. Please correct the equation.

We have corrected this mistake in the revised version.

p.5 l.26: is there a skin layer conductivity parameterization for the ocean ? If not it could be better to talk about diffusivity for the ocean instead of conductivity.

There is a skin layer parameterization for the ocean, details are provided in Beljaars (1997) and Zeng and Beljaars (2005). Thus, we judge it to be detailed enough in this description to give the surface energy budget equation in a general form, for details we refer to the IFS documentation.

 p.5 l.27-28: I don't understand this sentence. Does it mean that the albedo is prognostic ? The albedo is not a prognostic variable in ASCM, there it is provided as a boundary condition. In AOSCM, it is updated prognostic when sea-ice is present. It should indeed be upward coupling, so we changed the phrasing.

p.6. l.8-9: please add a reference about equation of state formulation

*We have added the references Fofonoff and Millard (1983) for EOS-80 and IOC et al. (2010) for TEOS-10.*

p.6 l.10: what is the interest to change the equation of state for the 1D ? numerical cost ?

To run the model based on TEOS-10, potential temperature and practical salinity need to be converted to conservative temperature and absolute salinity (which are conservative state variables). This approach does not notably increase the runtime. However, with EOS-80 less preparation of data is required and enhanced accuracy is not as important for the 1D model.

p.6 l.14: please give different time scale variable names for the different components (ocean, atmosphere, ice)

*Thank you for this suggestion. We have renamed the timescales tau\_a and tau\_o.*

p.6 l.20: please add Reffray et al. 2015 citation here.

The Reffray et al. (2015) reference is concerned with the PAPA mooring and other references exist in general about mixing parametrizations. We have included a more explicit reference to Reffray et al. (2015) at the beginning of Section 4.1.1.

p.8 l.34: can you give a practical example or reference about this statement please ? *We have added a reference for this statement.*

p. 9 l.14: can you give more details or practical recommendations/exemples about the relationship between the horizontal resolution of the host model and the SCM please ?

We added a sentence explaining "The resolution of the forcing is the main scale information applied in the model, apart from potential time-scale settings dependent on the horizontal grid settings".

p.9 l.24: the computation of the forcing data is not the same depending on the considered experiment and should consequently be moved in the corresponding experiment section.

The method applied to calculate the forcing information is the same for all experiments. Only the forcing frequency, vertical resolution and vertical extent (e.g. cut off above certain height for Arctic case) are specific to the three locations.

We have tried to highlight differences and similarities between simulations more clearly, also by introducing Table 1.

p.9 l.25: can you give explanations/practical reasons about the T511 resolution choice please? And the convective time step

The horizontal resolution is set to T511 to reduce instabilities, occurring due to a too long convective adjustment time scale, which can be found for example in the 2m temperature. The convective adjustment time scale is constant for grids finer than T511 and the choice of the horizontal resolution does not influence other parameters.

We have updated the sentence to read "to T511, mainly reducing the convective adjustment timescale and thereby alleviating instabilities."

p.9 l.25: "ASCM" acronym is not defined

Now we have defined both "ASCM" and "OSCM" in the general introduction (Sec 4.1).

p.9 l.31: specify that ORAS4 is a reanalysis *We have added this information.*

p.10 l.4: satellite chlorophyll climatology is used for Papa only or also in the 2 other locations ? If Papa only, this sentence can be moved in the 2.3.1 section.

Satellite chlorophyll data is used for all three locations. For increased clarity we have added in text " [...] climatologies are used. For the PAPA location the data is the same as presented in Reffray et al. (2015)."

p.10 l.21: please give the start date of the long simulation

The long simulations are started on the first of the respective month at 18 UTC. This was added to the manuscript.

**p.10 l.24-26: what about relative humidity?**

Observations of relative humidity have not been used in our study.

p.11 l.5: if all simulations are done with 60 vertical levels, this information can be moved in the general setup section.

Simulations are done on 60 levels for the PAPA and PIRATA locations but in the Arctic the number of levels is increased to 137. This information is now included in Table 1, which is describing the performed experiments.

**p.11 l.20: "loosely" ?**

We have removed the phrase "loosely", rewritten the sentence and also added the reference in which the complete setup is described at this point.

p.11 l.27: what is the LES boundary layer height ? is it constant ?

No, it is not constant. The vertical distribution of the forcing, and thus of the boundary layer, is shown in Figure 3 of Sotiropoulou et al. (2018).

p.11 l.28: why vertical advection from ERAI generates unrealistic results ? which kind of results ? please give more details about this point.

The vertical velocity is a difficult parameter to estimate from observations or models, as discussed in Section 2.2. In this particular case, we have observations that indicate that the cloud top height is decreasing in the beginning of the period and then remain constant at about 250 m. The vertical velocity from ERA-Interim is upward for some periods and results in a deeper cloud that is rising to about 1.5 km.

We think that this information does not fit the technical description paper of our model and can further be explained in a later paper.

p.12 l.6: what about the wind conditions associated with the cold advection event ?

Figure 7 shows that the period of weak advection is also associated with relatively weak winds. During the first four days of the considered period the winds at 10 m do not exceed 6 m/s and mostly stay slower than 4 m/s.

p.12 l.15: what can you conclude from the fact that results are similar between AOSCM3h and 6h? Please add a few comments.

We have added the following sentence: "During a period of weak atmospheric advection the frequency with which forcing information is updated is thus not influencing the evolution of the coupled column."

p.12 l.18-19: can you add in the text the local inertial period at Papa station to compare it with the simulation oscillations please ?

*Yes, this is a good idea. We have extended the text by: "At the location of the PAPA mooring the frequency of inertial oscillations is about 16 h."*

p.12 l.27: can you add the reanalysis in Figure 7 please to facilitate the comparison ? *See comment on Figure 7.*

p.12 l.30-31: you should add a figure showing this result.

In our study the focus is not on analysing physical phenomena but on demonstrating the model. Therefore, we choose to not present more detail on this result here, so we have added "(not shown)" at the end of this description.

p.12 l.31-32: the fact that nudging improves cloud and LW but deteriorate temperature and LH suggest there is errors compensation in your simulation. This should be stated in the text.

Not necessarily, a mismatch of observations and reanalysis (when the model is strongly nudged) can also indicate that local processes make it difficult to compare gridded and point information. We now mention both potential error sources in the text.

p.13 l.1: what about the skin SST parameterization in ERAI ?

ERA-Interim, as IFS in general, does include both a cool skin and a warm layer parametrizations. The same parametrisations are acting in OIFS cycle 40r1.

p.13 l.10: "sixteen" -> "five" ?!

The results shown in Tables 2 and 3 (new numbers) are mean and standard deviation of the RMSE from 16 29-day experiments which are done for 5 different setups (see new Table 1). The captions for tables 2-3 now include this information more clearly.

p.13 l.14: I think you have inverted "warm" and "cold" in this sentence.

Thank you for pointing out this mismatch. The reference to the months has been switched. Warm periods have a shallow mixed-layer and occur during June-September. p.13 l.15: "daily-mean" -> "observed daily-mean"

SST data used for ERA-Interim is not just based on observations but also including some post-processing, even data assimilation for some periods (Dee et al., 2011). These SST fields are also used here.

p.14 l.18-19: You cannot conclude that just by looking at the surface total heat flux in Fig. 8. A surface heat budget is needed for that.

We calculated the total surface energy budget (see equation (5) in the manuscript), which consists of the turbulent and radiative fluxes. We added the reference to the equation in the revision.

p.14 l.22-23: a timeseries with observed and simulated precipitations would be more convincing than Fig. 8b.

Accumulated precipitation reduces the noise compared to resolving the time series temporally. In addition, we are not interested if the timing and strength of each precipitation event is correctly modelled but just if the overall mass flux to the ocean is comparable.

p.15 l.34 – p.16 l.1: can you explain why the cloud formation is different from the LES ? Is it because the subsidence is not represented in the model ? If yes, why not force the model with a negative vertical velocity ?

The subsidence in the atmospheric component of the AOSCM is implemented in the same way as it is applied in the LES. The grid is coarser in the SCM, which gives a slightly different representation of the vertical distribution of heat and moisture, which gives a cloud with slightly more water in the AOSCM. Another difference is that whenever a cloud is formed in the LES, the region becomes turbulent due to radiative cooling. That process is not represented the same way in the IFS and may lead to subtle differences which then gives a different evolution. These detailed discussions are outside the scope of the present manuscript. They will be discussed in subsequent papers when the tool is used to answer questions on how the parameterisations interact and are able – or not – to capture observed evolutions.

p.16 l.10: the liquid water path is integrated over the entire atmosphere height ?

*Yes, the integrated LWP is calculated over the depth of the atmosphere, however, during the period only low-level clouds are contributing.*

p.16 l.22-23: is there any observations for the surface albedo ? if yes, can you validate your model albedo ? or directly use the observed albedo in your simulations ?

During ACSE, no albedo observations were made since that is not possible to do from a ship. Other sources of albedo for the location and time are discussed in Sotriopoulou et al. (2018). A longer discussion on this topic will be part of a later paper.

p.17 l.12: "infrequent": not sure if this the correct word to describe low-frequency forcing or the problems related to inertial frequency forcing

We changed the phrasing to be "temporally coarser data".

p.18 (should be page 17) l.26: you recommend pressure gradient forcing without testing it directly. It would be better to present it as a promising possibility that need to be tested.

We motivate pressure gradient forcing from a physical point of view on page 8 ll 15 (location in previous manuscript). However, it is correct that we do not present results from sensitivity experiments without pressure gradient forcing. We extend this sentence as "Based on the fluid dynamical theory and our results [...]".

\* Figures comments:

Figure 1: please detail the acronyms such as "GWD", : : :

Yes, this is a good suggestion. We have removed all acronyms in figures 1-3.

Figure 4: This schematic is confusing because LIM3 is a part of NEMO. Perhaps A big "NEMO" box with inside 2 small boxes such as "OPA" and "LIM3" would be easier to understand. *We have updated the figure as suggested.*

Table 2: a "Table 3" with oceanic RMSE would be interesting

Oceanic RMSE are very similar and not differing more than one standard deviation (calculated from 16 cases) between setups (5 different experiments). Therefore, we do not show this table. However, we now include a brief comment on this in Section 4.2.2.

Figure 6:

- Should be moved before Tables 1-2

*Yes, we moved two of the PAPA result figures ahead of the tables. Note that at this stage figures might not be in the correct order because latex is optimising page space.*

- A third panel showing data from ERAi and Papa mooring would greatly improve this figure. Information from ERA-Interim is included in the difference maps on the top row. The bottom

row includes ocean profiling information in the difference. Surface information from the PAPA mooring does not fit into this figure and is instead shown in Figure 7.

- Color bars are missing

This figure presents the values on the contour lines instead of through a colour bar, which would add more components to this complex figure.

- The initial mooring data appears to be missing in the figure, how do you initialize the ocean model if so ?

- Why are mooring oceanic observations missing between the surface and -10m ?

We cannot explain why ocean data is missing for some periods or height levels. However, some information is not missing but just appears to do (like initial profiles or surface values of salinity and temperature) because the contour plot does not resolve data if it is only available for one time step or one height level.

- Why did you chose this period if there are a lot of missing oceanic data ?

We choose this period because it is characterised by weak atmospheric advection. We do not want to investigate the origin of ocean biases further and therefore decided it is okay if data are not complete.

- BLH is computed from the AOSCM or ERAI ? Please clarify it.

- MDL is computed from observations or the AOSCM ? Please clarify it.

The BLH and MDL are computed in the AOSCM. We have added a clarification in the caption.

- (c) and (d) panels description are missing

Panel descriptions for (c) and (d) are included at the top of (a) and (b). We have added a note on this in the caption.

Figure 7: - please add Q2m and precipitation timeseries - please add ERAI to check how ERAI compare with observations and the model (it will also clarify your discussion).

This is a good idea in principal but most surface parameters are only available in the forecast fields and not in the 6-hourly ERA-Interim fields, from which the forcing was derived. The turbulent heat fluxes and radiation are model products and thus not suitable for evaluation of the AOSCM.

Figure 8: - please separate panels descriptions for (a) and (b) - legend is missing for grey squares - panel (b): wrong x axis: "fluxes" -> rain I would remove panel (b) and replace it by precipitation timeseries in Figure 6.

We have added separate labels for precipitation and now call it "Precipitation measured" and "Precipitation modelled". What seems to be grey squares are violet squares but with a brighter boundary instead of a dark blue boundary to indicate warm and cold points. Keeping the accumulated precipitation here simplifies the analysis compared to a noisy time series.

Figure 10 l.5: the red dots (cloud base) are not visible.

The figure caption was incorrect and we do not mention the red dots anymore.

Figure 12: the empty blue circle is not described in the legend. *Yes, it is described as "ASCM open blue symbol".*

- 30 (?)(Pithan et al., 2016). In the ocean, the depth of the mixed layer is sensitive to the coupling, especially in the tropics and during summer, when the mixed layer is shallow and quickly responding to forcing. The fast response can give rise to positive feedbacks between model biases in the atmospheric and oceanic mixed layers (Breugem et al., 2008; Toniazzo and Woolnough, 2014). It is common to develop model components using idealized prescribed forcing, i.e., ocean and land models use near-surface observed or reanalysis mean state variables to provide atmospheric fluxes. However, this can lead to surprises when
- 35 model components are interactively coupled. Atmospheric models are forced with observed SSTs over the ocean and often

[revised manuscript text omitted]

$$at5 - \dot{\eta}\frac{\partial u}{\partial \eta} + F_u + f(v - v_g) + \qquad P_u + \frac{u_r - u}{\tau}\frac{u_r - u}{\tau^a} = \frac{\partial u}{\partial t}$$
(1)

$$-\dot{\eta}\frac{\partial T}{\partial \eta} + F_T + \frac{RT\omega}{c_p p} + \qquad P_T + \frac{T_r - T}{\tau}\frac{T_r - T}{\tau_{am}} = \frac{\partial T}{\partial t}$$
(3)

10
$$-\dot{\eta}\frac{\partial q}{\partial\eta} + F_q + P_q + \frac{q_r - q}{\tau}\frac{q_r - q}{\tau} = \frac{\partial q}{\partial t}$$
 (4)

A As in the full model system, a two-time level semi-Lagrangian scheme is used (as in the full model system, an Eulerian scheme is optionalalso available) to integrate the momentum with horizontal wind components u and v (Eq. (1) and (2)), thermodynamics T (Eq. (3)), moisture q (Eq. (4)) as well as the continuity equation. The vertical coordinate is based on η levels, which merge orography following σ coordinates near the surface with pressure coordinates in the free atmosphere.
15 Here, ή and ω are vertical velocities, in η and pressure coordinates, respectively. Fi is the horizontal advection, Pi summarizes physical parametrizations and ur, vr, Tr, qr denote the reference profiles used for nudging with a time scale τTa. Furthermore, f is the Coriolis parameter, ug and vg the horizontal components of the geostrophic wind, R the moist air gas constant, cp the heat capacity of moist air at constant pressure and p the pressure. In addition to the atmospheric state variables (Eq. (1) - (4)), the model prognostically calculates cloud liquid, ice, rain, snow and cloud cover.

- The total tendency (right-hand sides of Eq. (1) Eq. (4)) to each prognostic variable is calculated as the sum of dynamical (first three terms on the left-hand side) and physical parametrization tendencies  $P_i$  (fourth term), possibly updated by relaxation (i.e. nudging, fifth term). The order of the left-hand side of the equation is, in a simplified way, equivalent to the sequence in which the tendencies are calculated in the model (Fig. 1). In the time-stepping loop, the dynamical tendencies are determined, mainly aggregating available prescribed forcing. The pressure gradient forcing is represented by the geostrophic wind. The
- 25 third term of the heat equation captures adiabatic heating through vertical motion. Calculations of tendencies from physical parametrizations are done in the same was as in the three-dimensional OIFS. Detailed discussion of the parametrizations used for these processes, namely, the radiation, turbulence, cloud and convection parametrization schemes as well as the non-orographic gravity wave drag, orographic gravity wave drag and surface drag, can be found in the IFS documentation for cycle 40r1 (https://www.ecmwf.int/sites/default/files/IFS CY40R1 Part4.pdf). Relaxation tendencies are calculated weighing

the difference between the new state, as determined by physical and dynamical tendencies, and a reference state, with the relaxation timescale  $\tau$ . All forcing is  $\tau_a$ . References states can, for example, be observed or modelled profiles of atmospheric variables. All forcing fields are read in at forcing time steps and linearly interpolated at intermediate model steps.

Besides visualising the sequence of main routines called during an OIFS SCM run, Fig. 1 also highlights in red communi-

5 cations with other AOSCM components through the coupler, and use of coupling variables in red. Coupling variables are also schematically shown in Figure 4. They enter the primitive equation system (Eq. (1) - (4)) via the surface energy balance budget (Eq. (5)).

$$(1 - \alpha_i)(1 - f_{Rs,i})R_s + \epsilon (R_T - \epsilon \sigma T_{sk,i}^4) + H_i + LH_i = Q_T = \Lambda_{sk,i}(T_{sk,i} - T_1)$$
(5)

The energy budget is solved individually for each surface tile *i*, which in the coupled system are the ocean and/or sea-ice. The 10 downward short-wave and long-wave radiations are  $R_s$  and  $R_T$ , with the tiled albedo  $\alpha_i$ , the surface tiled fraction of short-wave radiation absorbed at the surface  $f_{Rs,i}$ , the surface emissivity  $\epsilon$ , the Stefan-Boltzmann constant  $\sigma$ , the skin temperature  $T_{sk,i}$ , and the skin layer conductivity  $\Lambda_{sk,i}$ .  $H_i$  is the tiled sensible heat flux and  $LH_i$  the tiled latent heat flux. Downward Upward coupling is implemented through the surface albedo and the temperature of the upper snow, sea-ice or ocean layer  $T_1$ .

**2.1.2 NEMO**

15 NEMO is based on the thermodynamics and dynamics OPA model (Océan PArallélisé) and includes the LIM3 sea-ice component. More details of NEMO can be found in Madec (2016), and Rousset et al. (2015) describes the recent version of LIM. The ocean component NEMO3.6 is a primitive equation model based on the one-dimensional version of the Navier-Stokes equations (Eq. (6) and (7)), the hydrostatic equation, the incompressibility equation, heat and salt conservation equations (Eq. (8) and (9)), and the equation of state.

$$\quad at5 - \frac{\partial}{\partial z}\nu_t \frac{\partial u}{\partial z} + fv \qquad \qquad +P_u + \frac{u_r - u}{\tau} \frac{u_r - u}{\tau_o} = \frac{\partial u}{\partial t} \tag{6}$$

$$-\frac{\partial}{\partial z}\nu_t \frac{\partial v}{\partial z} - fu \qquad \qquad +P_v + \frac{v_r - v}{\underline{\tau}} \frac{v_r - v}{\underline{\tau}} = \frac{\partial v}{\partial t}$$
(7)

$$-\frac{\partial}{\partial z}K_t\frac{\partial T}{\partial z} + \frac{1}{\rho_o c_p}\frac{\partial I(F_{sol}, z)}{\partial z} + P_T + \frac{T_r - T}{\tau}\frac{T_r - T}{\tau_o} = \frac{\partial T}{\partial t}$$
(8)

$$-\frac{\partial}{\partial z}K_t\frac{\partial S}{\partial z} + E - P \qquad \qquad +P_S + \frac{S_r - S}{\tau}\frac{S_r - S}{\tau_0} = \frac{\partial S}{\partial t}$$
(9)

[revised manuscript text omitted]
., 2000). Another restriction is that some physical parametrizations can only be tested if the momentum balance is not artificially altered. While the cloud micro-physics description could be evaluated in a AOSCM applying relaxation, a Nudging momentum can be very helpful when evaluating cloud microphysics (e.g. Lohmann et al., 1999) but not in a study of the boundary-layer turbulence evolutionis not recommended. Nudging changes the equilibrium of
- 30 dynamic forcing and physical parametrizations, and might mask model biases. On the other hand, nudging tendencies can be evaluated and used to diagnose model drift and imbalances. Nudging is also useful as it allows handling of inaccurate or missing information, like inertial oscillations of wind or vertical velocity forcing.

[revised manuscript text omitted]

**4.1.1 Midlatitudes: PAPA station, east Pacific**

For the first experiment, we place the AOSCM at the PAPA mooring in the midlatitudinal east\_north-east Pacific (nominally at 50.1° N, 144.8° 50° N, 145° W, https://www.pmel.noaa.gov/ocs/Papa). Observations at this location have been extensively

30

used to develop physical parameterization parametrization in the ocean (e.g. Gaspar et al., 1990; Reffray et al., 2015), because the buoy is situated in a region of weak horizontal advection. Reffray et al. (2015) present a reference configuration of the NEMO column model at the PAPA mooring and test various mixing parametrizations available within NEMO.

The main experiment at the PAPA location consists of a 5-day coupled atmosphere-ocean simulation, initialized on 11 July 2014 at 18 UTC (11am local time) . The atmospheric model uses 60 levels and which is forced with 6-hourly data

(AOSCM-6h). The ocean column is resolved with 75 layers that are initialized from daily-mean profiles of temperature and salinity measured to a depth of 200 m. The coupling time step and coupling lag are chosen equal to the time step of the individual model components to be 900 s. Restart files of surface parameters required for coupled simulations are obtained from a short ASCM simulation. An uncoupled atmosphere-only simulations with 6-hourly atmospheric forcing (ASCM-6h)

- 5 and a coupled simulation with 3-hourly atmospheric forcing (AOSCM-3h) act as sensitivity runs to the main setup. One further set of simulations highlights how model drift in the free troposphere can be minimized. Here, nudging of temperature, moisture, and horizontal wind with a timescale of  $\tau = 6$ .  $\tau_a = 6$  h above a height of 3 km is applied (AOSCM-N3km6h). In addition, the model was run with the standard setting extended by relaxing the horizontal wind with a timescale of  $\tau = 1$  h  $\tau_a = 1$  h (AOSCM-Nuv0km1h). With each of the experiment settings described above, a further sixteen 29-day long simulations started
- 10 at 18 UTC on the first of the respective months (Oct 2010; Apr, Jun-Jul, Nov 2011; Mar, Aug, Nov 2012; Jun-Jul 2013; Jan, Apr, Jul-Sep, Nov 2014) are run for statistical assessment.

The atmospheric column of the model is first compared with ERA-Interim to ensure that the mean large-scale state is not drifting. Surface variables are evaluated using hourly averaged PAPA mooring surface measurements. The variables used here are, with measurement error estimates in parentheses: near-surface 2 m air temperature ( $\pm 0.2^{\circ}$  C), SST ( $\pm 0.003^{\circ}$  C), 10 m wind

15 speed ( $\pm 2\%$ ), wind-speed corrected precipitation ( $\pm 4 \text{ mm h}^{-1}$  on 10 min filtered data with measurement threshold of 0.2 mm h-1), long- and short-wave radiation (downwelling component with  $\pm 1\%$  error), and turbulent fluxes of heat. Thus, the model evaluation includes both a model-model comparison and an evaluation of grid-box mean results with point-measurements. The timeseries of ocean profiles at the PAPA location are influenced by tidal oscillations. As the model does not resolve these, the oscillations in measurements are removed by applying a running mean of 12h for the comparison.

**20 4.1.2 Tropical Atlantic**

The second marine-location at which the SCM is tested, lies in the tropical Atlantic, situated at the 6°S, 8°E buoy of the PIRATA mooring array (Servain et al., 1998; Bourlès et al., 2008, https://www.pmel.noaa.gov/tao/drupal/disdel/). We choose a boreal summer month to demonstrate the AOSCM's ability to follow the SST cooling connected to the annual cold tongue development in the topical\_tropical\_Atlantic (Lübbecke et al., 2010; Xie and Carton, 2004). During the period of 1-30 June

- 25 2014, mooring observations of SST, radiative fluxes, and ocean temperature and salinity are available for SCM evaluation. For evaluation of the experiments the observational data, which are complemented by ERA-Interim for the atmosphere. The ocean is initialised with daily mean temperature and salinity at the 1st of June from 0-500 m and 0-120 m respectively. We perform experiments in using several settings of the AOSCM and one OSCM simulation. The atmosphere is run with 60 levels and atmospheric column is either forced by advective tendencies and vertical velocity only (F01, F12, F15AOSCM-Jun1/12/15),
- 30 or additionally, profiles of temperature, moisture, and horizontal wind are nudged above 1 km with a timescale of 6 hours (N1km6h). For comparison AOSCM-N1km6h). For comparison, we also perform an ocean-only simulations (OSCM), which is forced by hourly precipitation, near-surface wind, temperature and moisture from ERA-Interim, and shortwave- and longwave radiation measured at the PIRATA buoy(OSCM). The oceanic, atmospheric, and coupling time steps are 900 s each.

**4.1.3 North Polar region**

To explore the AOSCM in an experimental setting with idealized forcing, and to show the additional interaction with sea ice, we choose an Arctic summer case. For this location (76° N, 160° E), we have observations from the ACSE (Arctic Clouds in Summer Experiment) campaign during a warm-air advection episode in early August 2014 causing rapid ice melt (Tjernström

- 5 et al., 2015). Sotiropoulou et al. (2018) apply idealized forcing to a LES case use an LES to study the importance of advection for cloud evolution during this period. Here, we present results from the LES (Savre et al., 2014), in comparison with results from the ASCM, using the same experimental setup as in Sotiropoulou et al. (2018). Furthermore, we explore the importance of coupling to the ocean/sea ice, as well as the sensitivity to atmospheric model time step and coupling frequency, in ASCM and AOSCM experiments. With the aim to separate the influence of local and remote processes, as in Sotiropoulou et al. (2018),
- 10 we turn off large-scale advection of heat and moisture.

The idealized experiment, loosely based on observations based on simplified information from observations and reanalysis (Sotiropoulou et al., 2018), assumes an initial ice concentration (100 %), surface albedo (0.65), and temperature (273.15 K, i.e. melting point of ice). The LES is applying a surface friction velocity of  $\frac{0.2}{0.2} u_* = 0.2$  m s-1 as lower boundary condition, while it is modelled in the ASCM and AOSCM using a surface roughness, updated from its default value (0.001 m) to 0.06

- 15 m to achieve approximately the same averaged  $u_*$ . The LES and the atmospheric component of AOSCM are initialized with the same vertical mean profiles, smoothed versions of soundings at 1 August 06 UTC, the starting time of the simulation. The atmospheric forcing consists of a constant geostrophic wind of 5.4 m s-1 and advective tendencies of temperature and humidity, all derived from 6-hourly ERA-Interim data interpolated to a vertical L137 grid, but restricted vertically to the LES boundary layer height. All forcing is 6-hourly in accordance with the LES setup. The synoptic scale divergence (i.e. vertical
- advection), is not directly taken from the ERA-Interim as it generates unrealistic results. Thus, a prescribed divergence of  $2.3 \cdot 10^{-5}$  s-1 is applied over the first 18 simulated hours and then decreased by 50 %, in both the LES and the SCM experiments.

**5 Results**

**4.1 **Results from experiments**

**25 4.2 PAPA mooring**

As a first marine test location we choose the PAPA mooring in the east Pacific. Results are presented from one short summer case study and from 16 month-long simulations distributed throughout the year.

**4.1.1 Case study**

[revised manuscript text omitted]

OIFS master

---

## Author Comment (AC3) · 24 Aug 2018

The manuscript provides a detailed description of a coupled atmosphere – sea ice – ocean column model and examples of its applications. The model components and the coupling methods are basically well presented. Particularly good in the manuscript is the discussion on how to design experiments applying the coupled column model. Also the Introduction is very good and motivates a reader. The manuscript has potential to become a good paper, and particularly useful for those who carry out column model experiments or plan to start such activities. The manuscript has, however, also weaknesses, and substantial revisions are needed before I can recommend it for publication.

*We would like to thank the anonymous reviewer for the useful feedback. We have corrected typos and adapted suggestions for textual changes. The changes may easily be followed in the attached document containing track changes.*
*For suggestions requiring more explanations, we provide answers below.*

Major comments
1. In several parts of the manuscript, the text is too technical, including acronyms never explained, e.g. on page 6, lines 8, 10, and 17; page 7, lines 12, 14, and 15, and many other places. This holds also for Figures 1 to 3, which are more like computer codes than illustrations that one would expect to see in a peer-reviewed journal.
    *Yes, it is indeed true that some abbreviations and technical terms were not sufficiently explained in the first version of the manuscript. We wanted to include technical terms as our article should also be useful to those who will be working with the model code and its development. We now tried to give generally understandable explanations but also refer to the technical terms.*
    *P 6: We no longer include the technical terms of the equations of state in the text (l. 6 and 8) and have also added the name of the time stepping routine in brackets.*
    *P 7: The special technical terms concerning the OASIS routines are now also included in brackets.*
    *Figures 1-3: When we started documenting the model, we came across the GMD publication https://www.geosci-model-dev.net/10/1549/2017/ which uses code illustrations to highlight the work done. We think this visualisation is useful but we have tried to clarify and reduce the content of the tree schematics.*

2. The description of sea ice component is very brief, and leaves many open questions. How many model levels there are in sea ice and snow? How the grid is affected when ice and snow thicknesses change, i.e., is the number of levels changed or is the vertical resolution changed? Is penetration of solar radiation into snow and ice taken into account (as in case of the open ocean)? Is snow-to-ice transformation (via refreezing of melt water, rain, or sea water flooded on top of ice) taken into account? Are melt ponds taken into account? These are important questions in atmosphere – ocean coupling in Polar regions. Figure 3 does not provide any help on these issues.
    *We now give a bit more information about the setup of the sea-ice and snow component (2 vertical layers, thickness categories which are constant in time and only the distribution of fractions is changing).*

*However, we decided to not go into too much detail in our paper but refer to the respective modelling site. This is partly due to the fact that modelling details change all the time and we want to keep our paper from being outdated soon. We now explain briefly in the model overview section (2.1) why the model descriptions are not very detailed.*

3. I am concerned about conclusions made on the basis of comparisons of the model results and ERA-Interim reanalyses in the open-ocean experiments. Are the vertical profiles in ERA-Interim over remote ocean areas accurate enough to make robust conclusions on the performance of the column model? Some references or other evidence on the accuracy of the ERA-Interim profiles should be provided. Further, on page 11, line 3, it is stated that observational data are complemented by ERA-Interim for the atmosphere. It should be more clearly explained what atmospheric variables were observed.

*This is a very relevant comment. ERA-Interim is, in the first place, used to initialise and force the atmospheric column. We decided to also use the reanalysis profiles as a test for model drift. Although updated by data assimilation, ERA-Interim is based on the same atmospheric model as the AOSCM. Thus, the AOSCM should follow the forcing profiles approximately. We have added a brief discussion of this topic in Section 4.1.*
*Comparison with observations is necessary for a strict evaluation of the AOSCM. These data were available near the surface, where the newly coupled model should produce the most benefits.*
*It is definitely desirable to extent the assessment of model performance, however, this should be content of following publications.*

*Observations used for analysis at the PIRATA mooring were presented on page 11, line 3 (old location). Namely, SST, radiative fluxes as well as profiles of ocean temperature and salinity. We have kept this information.*

4. The presentation of the results of the three cases studies should be clarified. It is presently hard to understand the basis of the conclusions made.

*The main goal of the results section is to demonstrate the versatility of the AOSCM, while at the same time discussing different types of experimental setups as well as testing the model's stability and sensitivities to some of the model parameters.*
*We have restructured the text to make it clearer that the experiments section belong together. Conclusions are presented in an overview section on model results (Section 4.3). Additional results to scientific questions are not the main part of our study and therefore not explained in detail.*

5. On page 11, line 24, it is told that 0.06 m was used for sea ice roughness. This value is much larger than observations indicate (see papers by Andreas, Lupkes and others). If the key parameter for dynamical coupling between the atmosphere and sea ice is so strongly tuned, what is the relevance of the column model experiments?

*Roughness length was here set to achieve an agreement of the friction velocity with LES simulations and observations, to be able to compare with the LES.*
*In models, the relationship between surface roughness and the strength of the turbulence (i.e. the friction velocity) usually depends on the vertical stability. The momentum coupling is known to be difficult and it is generally not possible to reproduce all aspects of the atmospheric state associated to this coupling, e.g. the boundary layer wind-turning, the jet speed and frequency of boundary layer decoupling from the surface. See for example Sandu et al. (2013).*
*This might explain why a different roughness length is necessary in the model in comparison to observations and the LES.*

6. Advective tendencies were not applied in the ocean, but could be obtained from ocean reanalyses, analogously to obtaining them from ERA-Interim.

*Yes, this is correct. Currently the ocean model does not allow forcing with advective tendencies. It is planned to include this in later versions of the AOSCM.*
*However, although it is technically possible to calculate the ocean advection, ocean reanalyses fields are often only available as monthly means or as running averages over a few days. It would probably be necessary to obtain ocean advection from global climate model results.*

7. In Summary and Outlook the authors should better evaluate which processes can be simulated applying AOSCM and which cannot. Considering processes acting in the vertical dimension in the Arctic Ocean, there are major challenges in understanding and modelling at least the following processes: gravity drainage of salt in sea ice, brine formation, turbulent exchange of momentum, heat and salt at the ice base during ice growth, double-diffusive convection, and those mentioned in comment 2 above. After reading the manuscript, it remains unclear if AOSCM takes these processes into account.

*Our paper focuses on presenting the coupled atmosphere-ocean single column model and its general way of use. All thermodynamic processes described by vertical parametrizations can be studied with the AOSCM. Details on physical parametrizations of the different sub-components cannot all be included, but we refer to the respective model development literature.*
*Please also see answer to comment 2.*

Minor comments
Page 3, line 7: Do you mean 44% of meteorological modelling centres or modelling centres in general?

*It is modelling centres which develop coupled atmosphere and ocean models. We have clarified this in the text.*

Page 5, line 18: Clarify what is meant by a reference state.

*We now explain the term "reference state" both in Section 2.1.1 and Section 2.2.*

Equations (5) and (8): If part of the solar radiation penetrates below the surface, as in Eq. (8), it should not contribute to surface energy balance, but this is not reflected in Eq. (5).

*We forgot to include one parameter, which describes the fraction of solar radiation penetrating below the skin layer. Now the surface energy budget is consistent with penetration of solar radiation into the ocean and sea-ice.*

Page 7, line 21: does "fraction" mean areal coverage of sea ice?

*This is correct, we have included a comment in the revision.*

Page 9, line 12: Briefly explain what is asynchronous coupling.

*Thanks for this suggestion. We now briefly explain an alternative coupling option. Also, there was a mistake before and the more advanced coupling is synchronous coupling.*
*"An example of a more advanced coupling method is synchronous coupling (Lemarie et al., 2015) in which coupling fields are sent and received at the same time."*

Page 10, line 24: Does "near-surface temperature" mean 2-m air temperature?

*Yes, this is correct. We changed the phrasing in the text.*

Page 10, line 28: Was the tide only due to M2 component? Is the diurnal tide had large contribution, 12 h running means do not help. Note that tidal currents may also affect vertical mixing in the ocean (although not necessarily much in the case presented, as it is in the deep ocean) and, if the model does not include tides, temporal averaging does not help.

*The vertical displacement of density surfaces due to tides is not captured in the model. However, mixing resulting due to internal tide breaking is parametrized.*
*Before we applied the running mean over the observations of salinity and temperature we checked for the main frequency peak in the oscillations. The data exhibit a maximum at around 12 h and no peak is visible at around 24 h. We have added this information to the manuscript.*

Page 12, lines 29-31: Why did the results improve?
    *We did not follow up what physical processes lead to reduced biases. This could be the content of a follow up paper and does not fit the model description paper.*

Page 14, line 31: Why not simply write RMSE instead of confusing "integrated bias".
    *Yes, this is a good suggestion.*

Page 17, line 2: Why do you call EC-Earth as "the next version climate model"? It sounds very strange, particularly when referring to a paper eight years old.
    *Yes, this is a good comment. The paper refers to EC-Earth version 2. Our AOSCM is similar to version 3, apart from using OIFS instead of IFS. The plan is that for EC-Earth version 4 OIFS will be used as the atmospheric model so that similarities between the 3D and 1D models increase. We removed the reference to the version 2 paper at this point.*

[revised manuscript text omitted]